# Scalable Power Sampling: Unlocking Efficient, Training-Free Reasoning for LLMs via Distribution Sharpening

Xiaotong Ji [* 1]   Rasul Tutunov [* 1]   Matthieu Zimmer [1]   Haitham Bou Ammar [1 2]

## Abstract

Reinforcement learning (RL) post-training is a dominant approach for improving the reasoning performance of large language models (LLMs), yet growing evidence suggests that its gains arise primarily from distribution sharpening rather than the acquisition of new capabilities. Recent work has shown that sampling from the power distribution of LLMs using Markov chain Monte Carlo (MCMC) can recover performance comparable to RL post-training without relying on external rewards; however, the high computational cost of MCMC makes such approaches impractical for widespread adoption. In this work, we propose a theoretically grounded alternative that eliminates the need for iterative MCMC. We derive a token-level decomposition of the autoregressive power distribution, showing that the global power distribution can be approximated by a token-level scaled low-temperature one, where the scaling factor captures future trajectory quality. Leveraging this insight, we introduce a training-free and verifier-free algorithm that sharpens the base model's generative distribution autoregressively. Empirically, we evaluate our method on math, QA, and code tasks, and show that our method matches or surpasses one-shot GRPO *without* relying on any external rewards, while reducing inference latency by over $10\times$ compared to MCMC-based sampling.

## 1. Introduction

The current frontier of large language models (LLMs) reasoning is largely defined by reinforcement learning (RL) post-training. By optimising models against automated veri-fiers, such as unit tests in coding tasks (Chen et al., 2021; Roziere et al., 2023; Hou et al., 2024), final-answer checks or Lean certificates in mathematics (Cobbe et al., 2021; Yang et al., 2023; Ahn et al., 2024; Xin et al., 2024; Zimmer et al., 2025), methods like Group Relative Policy Optimisa-tion (GRPO) and its variants have established state-of-the-art performance across a range of benchmarks (Zhao et al., 2023; Shao et al., 2024; Yu et al., 2025). However, the mechanisms underlying these gains remain the subject of active debate. A growing body of evidence suggests that RL may not introduce fundamentally new reasoning capabilities, but instead acts as a form of distribution sharpening (He et al., 2025; Song et al., 2025; Yue et al., 2025; Gai et al., 2025; Ni et al., 2025). Under this view, the reasoning trajectories required to solve complex problems already exist within the base model; RL primarily reshapes the probability mass.

If post-trained models are indeed "sharpened" versions of the base model, it follows that comparable reasoning capabilities should be attainable by directly targeting a sharpened distribution of the base LLM without relying on external validation rewards. Recent work by Karan & Du (2025) provides compelling evidence for this view. They make use of the power distribution, $p^\alpha(x)$, as a principled mechanism for inducing such sharpening without any additional training. Strikingly, their results show that sampling from the power distribution enables base models to match and, in some cases, surpass the performance of RL-trained models on challenging benchmarks, such as MATH500 and GPQA. These findings strongly support the hypothesis that the reasoning capabilities attributed to RL are largely latent within the base model and can be revealed through an appropriate sampling strategy.

Although the power distribution offers a compelling mechanism for sharpening base model capabilities, current methods for sampling from it are computationally prohibitive. Karan & Du (2025) argue that standard autoregressive sampling methods like those in low-temperature sampling (Wang et al., 2020) fail to capture the global properties of $p^\alpha$. Consequently, they resort to MCMC techniques, specifically a Metropolis-Hastings algorithm that iteratively resamples a token subsequence. While effective, this approach introduces significant latency, requiring multiple forward passes

---

[*]Equal contribution [1]Huawei Noah's Ark Lab [2]UCL Centre for Artificial Intelligence. Correspondence to: Haitham Bou Ammar <haitham.ammar@huawei.com>.

*Proceedings of the $43^{rd}$ International Conference on Machine Learning*, Seoul, South Korea. PMLR 306, 2026. Copyright 2026 by the author(s).

and iterative refinement steps to generate a single response. Such computational overhead restricts the method's scalability for real-world deployment.

To enable broad adoption, we seek a faster and more scalable sampling procedure while retaining the performance advantages of MCMC. To this end, we revisit the fundamental relationship between the power distribution and the low-temperature distribution. While prior work has established that these distributions are theoretically distinct, we derive a novel formulation showing that the power distribution can be effectively approximated by a suitably scaled low-temperature policy. We explicitly identify the required scaling factor and theoretically quantify the accuracy of this approximation. This insight allows us to bypass the iterative cost of MCMC entirely: by proposing a (*training-free and verifier-free*) algorithm that applies this scaling during standard autoregressive generation, we efficiently target the sharpened distribution, retaining the reasoning benefits of the power distribution while reducing inference latency by over 10×. In short, our contributions can be summarised as:

(1) We bridge the theoretical gap between global power distributions and local autoregressive sampling. We prove that the power distribution can be decomposed into a scaled low-temperature one, deriving a closed-form scaling factor and providing rigorous bounds on the approximation error.

(2) We introduce a training- and verifier-free algorithm that enables direct autoregressive generation from the sharpened distribution. This eliminates the iterative overhead of MCMC, reducing inference latency by over 10× while preserving the reasoning improvements of the global power distribution.

(3) Across three benchmarks (MATH500, HumanEval, GPQA), we show that our method improves over strong reward-free inference baselines, and matches or surpasses one-shot GRPO performance in several in-domain and out-of-domain settings, without any parameter updates or external rewards.

## 2. Power & Low-Temperature Distributions

In prior work, Karan & Du (2025) show that the power distribution is fundamentally distinct from standard low-temperature sampling. We formalise both distributions here and discuss those challenges. Let $p(\cdot|\boldsymbol{q})$ denote the conditional probability distribution induced by a pretrained language model on sequences from a finite vocabulary $\mathcal{V}$ conditioned on an input prompt $\boldsymbol{q}$. Given a parameter $\alpha > 1$ and a partially generated token sequence $\boldsymbol{x}_{0:t-1} = \{x_1, \ldots, x_{t-1}\}$, we define the low-temperature (Wang et al., 2020) and the associated power distributions

for the next token $x_t$ as follows:

$$p_\alpha^{(\text{low. temp})}(x_t|\boldsymbol{q}, \boldsymbol{x}_{0:t-1}) = \frac{p^\alpha(x_t|\boldsymbol{q}, \boldsymbol{x}_{0:t-1})}{\sum_{x_t' \in \mathcal{V}} p^\alpha(x_t'|\boldsymbol{q}, \boldsymbol{x}_{0:t-1})}, \quad (1)$$

$$p_\alpha^{(\text{pow})}(x_t|\boldsymbol{q}, \boldsymbol{x}_{0:t-1}) = \frac{\sum_{\boldsymbol{x}_{t+1:T}} p^\alpha(\boldsymbol{q}, \boldsymbol{x}_{0:t-1}, x_t, \boldsymbol{x}_{t+1:T})}{\sum_{\boldsymbol{x}_{t:T}} p^\alpha(\boldsymbol{q}, \boldsymbol{x}_{0:t-1}, \boldsymbol{x}_{t:T})},$$

Equation 1 shows that sampling from the low-temperature distribution, $p_\alpha^{(\text{low. temp})}$, is relatively straightforward, as it depends only on the current history and token-level probabilities. In contrast, sampling from the power distribution is substantially more challenging, since it requires marginalising over all possible future completions in both the numerator and denominator of $p_\alpha^{(\text{pow})}$:

$$p_\alpha^{(\text{pow})}(x_t \mid \boldsymbol{q}, \boldsymbol{x}_{0:t-1}) = \frac{\sum_{\boldsymbol{x}_{t+1:T}} p^\alpha(\boldsymbol{q}, \boldsymbol{x}_{0:t-1}, x_t, \boldsymbol{x}_{t+1:T})}{\sum_{\boldsymbol{x}_{t:T}} p^\alpha(\boldsymbol{q}, \boldsymbol{x}_{0:t-1}, \boldsymbol{x}_{t:T})},$$

which we colour-coded in blue and green in the above equation for clarity. Of course, those marginalisations require summing over all possible future token sequences, whose cardinality grows exponentially with the remaining horizon $T - t$, i.e., $|\mathcal{V}|^{T-t}$ for a vocabulary $\mathcal{V}$. This intractability makes direct sampling impractical in general and motivates the need for approximate inference procedures.

It also highlights that the low-temperature and power distributions are *fundamentally different*: whereas low-temperature sampling applies a local transformation at each token, the power distribution induces a global, trajectory-level reweighting. In fact, Karan & Du (2025) show that these two distributions generally do not coincide and can differ at every token position.

**Approximate Sampling from $p_\alpha^{(\text{pow})}$.** To approximately sample from this sharper power distribution, Karan & Du (2025) resort to approximate inference via Markov chain Monte Carlo (MCMC). They employ a Metropolis–Hastings scheme that constructs a Markov chain over full token sequences, using the base model likelihoods to define acceptance ratios. Concretely, their method iteratively proposes partial resampling of token subsequences and accepts or rejects these proposals to asymptotically target the power distribution. To improve practicality in the autoregressive setting, they further introduce a variant that samples from a sequence of intermediate, prefix-level power distributions, gradually extending the sequence while running a bounded number of MCMC steps at each stage.

This approach inherits MCMC's classical high-dimensional limitations, where mixing times can be exponential (Levin & Peres, 2017; Gheissari et al., 2018). Practically, this causes significant overhead: Karan & Du (2025) reports an approximately **8.8× cost increase** over standard sampling for MATH500 sequences. Motivated by this, we derive an

efficient approximation via a new principled connection between low-temperature sampling and the power distribution. This eliminates iterative MCMC loops whilst preserving future-aware reweighting behaviours.

## 3. Approximating Power Distributions

We now turn to the central technical contribution of this work. Although prior results (Karan & Du, 2025) establish that low-temperature sampling generally does not recover the power distribution, we show that the power distribution nevertheless admits a structured token-level decomposition that directly relates it to low-temperature sampling. This observation allows us to move beyond a binary distinction between the two and instead characterise their relationship explicitly. In particular, we prove that the power distribution can be expressed as a scaled version of the low-temperature distribution $p_\alpha^{(\text{low. temp})}$. This result provides a principled foundation for approximating power-distribution sampling using standard autoregressive procedures, and forms the basis of the efficient inference methods developed in the remainder of this section.

The following theorem makes this decomposition explicit by providing a closed-form expression for the scaling term that relates the power distribution to the low-temperature counterpart at each token position:

**Theorem 3.1.** *Let $p(x)$ be a pretrained large language model, $\boldsymbol{q}$ an input prompt and $\alpha > 1$ be an exponent parameter. Then, for any partially generated sequence $\boldsymbol{x}_{0:t-1} = \{x_0, \ldots, x_{t-1}\}$ we have:*

$$p_\alpha^{(pow)}(x_t|\boldsymbol{q}, \boldsymbol{x}_{0:t-1}) = \frac{p^\alpha(x_t|\boldsymbol{q}, \boldsymbol{x}_{0:t-1})\zeta_t(x_t)}{\sum_{x_t' \in \mathcal{V}} p^\alpha(x_t'|\boldsymbol{q}, \boldsymbol{x}_{0:t-1})\zeta_t(x_t')},$$
(2)

*where for each token $x_t' \in \mathcal{V}$, we have that: $\zeta_t(x_t') = \sum_{\boldsymbol{x}_{t+1:T}} p^\alpha(\boldsymbol{x}_{t+1:T}|\boldsymbol{q}, \boldsymbol{x}_{0:t-1}, x_t')$.*

*Proof sketch.* For a fixed prefix $(\boldsymbol{q}, \boldsymbol{x}_{0:t-1})$, the next-token conditional under the sequence-level power distribution marginalises over all future completions. Applying the chain rule factors each powered joint likelihood into the prefix likelihood, the powered next-token probability, and a future-completion term; the prefix likelihood cancels between numerator and denominator. The remaining summed future-completion term gives $\zeta_t(x_t)$, and normalising over candidate next tokens yields Equation 2; see Appendix A.1 for the full derivation. □

Theorem 3.1 reveals that the difference between power-distribution sampling and low-temperature sampling is entirely captured by a token-specific scaling factor that reflects the expected likelihood of future completions. While low-temperature sampling favours tokens that are locally probable under the model, the power distribution further reweights these tokens according to the average quality of the sequences they induce downstream. In this sense, power sampling can be viewed as low-temperature sampling augmented with a future-aware correction. Importantly, this decomposition localises the global, trajectory-level dependence of the power distribution into a per-token scalar term, explaining both why MCMC-based methods were previously required and how they can be avoided. As the generation progresses and the remaining horizon shortens, the power distribution naturally approaches the low-temperature one, where the two exactly coincide on the final token $x_T$:

$$p_\alpha^{(\text{pow})}(x_T|\boldsymbol{q}, \boldsymbol{x}_{0:T-1}) = p_\alpha^{(\text{low. temp})}(x_T|\boldsymbol{q}, \boldsymbol{x}_{0:T-1}).$$

### 3.1. Autoregressive Monte Carlo Estimation

The key insight of Theorem 3.1 is that power-distribution sampling can be performed autoregressively, provided that each token probability is modulated by an appropriate token-wise scaling factor $\zeta_t(x_t')$. Crucially, this scaling factor admits an expectation form under the *base language model*, which allows it to be estimated via *standard autoregressive Monte Carlo sampling*. As a result, the power distribution can be efficiently approximated without resorting to trajectory-level inference such as MCMC.

Formally, for any candidate token $x_t' \in \mathcal{V}$, the scaling factor can be rewritten as an expectation with respect to the base language model:

$$\zeta_t(x_t') = \mathbb{E}_{\boldsymbol{x}_{t+1:T} \sim p_f(\cdot)}\left[p^{\alpha-1}(\boldsymbol{x}_{t+1:T}|\boldsymbol{q}, \boldsymbol{x}_{0:t-1}, x_t')\right],$$

with $p_f(\cdot) \equiv p(\boldsymbol{x}_{t+1:T}|\boldsymbol{q}, \boldsymbol{x}_{0:t-1}, x_t')$ denoting the base language model's distribution over future completions. Thus, a Monte Carlo approximation of $\zeta_t(x_t')$ can be easily attained as follows: *i)* generate $M_t$ i.i.d. completions $\{\boldsymbol{x}_{t+1:T}^{[r]}\}_{r=1}^{M_t}$ from the LLM when conditioned on $\{\boldsymbol{q}, \boldsymbol{x}_{0:t-1}, x_t'\}$, and *ii)* compute an empirical estimate of $\zeta_t(x_t')$ as:

$$\hat{\zeta}_t(x_t') = \frac{1}{M_t}\sum_{r=1}^{M_t} p^{\alpha-1}(\boldsymbol{x}_{t+1:T}^{[r]}|\boldsymbol{q}, \boldsymbol{x}_{0:t-1}, x_t'). \quad (3)$$

Substituting those Monte Carlo estimates into the decomposition of Theorem 3.1 yields the following approximation to the power distribution:

$$\hat{p}_\alpha^{(\text{pow})}(x_t \mid \boldsymbol{q}, \boldsymbol{x}_{0:t-1}) = \frac{p^\alpha(x_t \mid \boldsymbol{q}, \boldsymbol{x}_{0:t-1})\,\hat{\zeta}_t(x_t)}{\sum_{x_t' \in \mathcal{V}} p^\alpha(x_t' \mid \boldsymbol{q}, \boldsymbol{x}_{0:t-1})\,\hat{\zeta}_t(x_t')},$$
(4)

with $\hat{\zeta}_t(x_t)$ denoting the scaling factor at the token $x_t$.

### 3.2. Bias Analysis and Correction

Although each scaling factor is estimated without bias using $\hat{\zeta}(x_t)$ and $\hat{\zeta}(x_t')$, the substitution step back in Theorem

3.1 to form $\hat{p}_\alpha^{(\text{pow})}(\cdot)$ induces bias in the resulting estimator, since the expectation of a ratio does not equal the ratio of expectations. Although one could directly use this estimator in practice, we instead explicitly analyse the source of this bias and introduce a jackknife-based correction that significantly reduces it while preserving the efficiency of the method.

The following Lemma shows that the bias of the estimator $\hat{p}_\alpha^{(\text{pow})}(x_t|\boldsymbol{q}, \boldsymbol{x}_{0:t-1})$ decays asymptotically at a rate inversely proportional to the sampling budget $M_t$:

**Lemma 3.2.** *Let $\hat{p}_\alpha^{(\text{pow})}(\cdot|\boldsymbol{q}, \boldsymbol{x}_{0:t-1})$ be the Monte Carlo estimator of the power distribution defined in Equation 4. Then, we have:*

$$\mathbb{E}\left[\hat{p}_\alpha^{(pow)}(\cdot|\boldsymbol{q}, \boldsymbol{x}_{0:t-1})\right] = p_\alpha^{(pow)}(\cdot|\boldsymbol{q}, \boldsymbol{x}_{0:t-1}) + \mathcal{O}\left(\frac{1}{M_t}\right),$$

*where the expectation $\mathbb{E}[\cdot]$ is taken with respect to the random completions used to estimate the scaling factors $\zeta_t(x_t')$ for all tokens $x_t' \in \mathcal{V}$.*

From the above, we see that the bias can be reduced by increasing the number of sampled trajectories $M_t$ from the base language model. However, since each new sample requires additional autoregressive generation, this approach quickly becomes computationally expensive. We therefore seek an alternative that reduces the bias more rapidly with respect to $M_t$, enabling accurate estimation with substantially fewer samples from the LLM. The proof is provided in Appendix A.2.

**Jackknife Estimator for Bias Reduction.**  To do so, we adopt a jackknife correction. The jackknife is a classical bias-reduction technique that constructs corrected estimators by systematically recombining leave-one-out versions of an original estimator (Miller, 1974; McIntosh, 2016). When applied to estimators that depend smoothly on empirical averages, the jackknife removes the leading-order bias term while preserving the overall computational structure. This makes it particularly well-suited to our setting, where the bias arises from taking ratios of Monte Carlo estimates.

The key idea is to construct a bias-reduced estimator by combining the original Monte Carlo approximation with a collection of *leave-one-out variants*, each obtained by removing a single sampled trajectory. This construction yields the following jackknife-corrected estimator of the power distribution:

$$\hat{p}_{\alpha,\,\text{JK}}^{(\text{pow})}(x_t|\boldsymbol{q}, \boldsymbol{x}_{0:t-1}) = M_t\hat{p}_\alpha^{(\text{pow})}(x_t|\boldsymbol{q}, \boldsymbol{x}_{0:t-1}) \quad (5)$$
$$- \frac{M_t - 1}{M_t}\sum_{s=1}^{M_t}\hat{p}_{\alpha,-s}^{(\text{pow})}(x_t),$$

where $\hat{p}_{\alpha,-s}^{(\text{pow})}(x_t)$ denotes the leave-one-out Monte Carlo estimator obtained by excluding the $s^{\text{th}}$ trajectory. Here, we

define the leave-one-out-estimation of $\zeta_t(x_t')$ as:

$$\hat{\zeta}_{t,-s}^{(\text{LOO})}(x_t') = \frac{1}{M_t - 1}\sum_{r=1,r\neq s}^{M_t} p^{\alpha-1}(\boldsymbol{x}_{t+1:T}^{[r]}|\boldsymbol{q}, \boldsymbol{x}_{0:t-1}, x_t'),$$

(6)

This, in turn, leads us to the following definition of the leave-one-out approximation of the power distribution:

$$\hat{p}_{\alpha,-s}^{(\text{pow})}(x_t) = \frac{p^\alpha(x_t|\boldsymbol{q}, \boldsymbol{x}_{0:t-1})\hat{\zeta}_{t,-s}^{(\text{LOO})}(x_t)}{\sum_{x_t'\in\mathcal{V}} p^\alpha(x_t'|\boldsymbol{q}, \boldsymbol{x}_{0:t-1})\hat{\zeta}_{t,-s}^{(\text{LOO})}(x_t')},$$

which when substituted back in Equation 5 gives us the final Jackknife estimator $\hat{p}_{\alpha,\,\text{JK}}^{(\text{pow})}(x_t|\boldsymbol{q}, \boldsymbol{x}_{0:t-1})$.

The following lemma formalises the benefit of this construction by showing that the jackknife estimator cancels the leading-order bias term, yielding a bias that decays at an inverse-quadratic rate in $M_t$:

**Lemma 3.3.** *Let $\hat{p}_{\alpha,\,\text{JK}}^{(\text{pow})}(\cdot|\boldsymbol{q}, \boldsymbol{x}_{0:t-1})$ be the jackknife estimator of the power distribution defined in Equation 5. Then*

$$\mathbb{E}\left[\hat{p}_{\alpha,\,JK}^{(pow)}(\cdot|\boldsymbol{q}, \boldsymbol{x}_{0:t-1})\right] = p_\alpha^{(pow)}(\cdot|\boldsymbol{q}, \boldsymbol{x}_{0:t-1}) + \mathcal{O}\left(\frac{1}{M_t^2}\right),$$

This result has an important computational implication. For the original Monte Carlo estimator without jackknife correction, achieving a bias of at most $\epsilon$ requires $M_t = \mathcal{O}(1/\epsilon)$ sampled trajectories. In contrast, the inverse-quadratic bias decay established above implies that the jackknife estimator attains the same bias level with only $\mathcal{O}(1/\sqrt{\epsilon})$ samples. Thus, the jackknife correction substantially reduces the number of required language-model generations for a given accuracy. The proof is provided in Appendix A.3.

It is easy to see that an immediate corollary of Lemma 3.3 is convergence in expectation of $\hat{p}_{\alpha,\,\text{JK}}^{(\text{pow})}(\cdot)$ to the true power distribution $p_\alpha^{(\text{pow})}(\cdot)$ as the number of Monte Carlo trajectories $M_t \to \infty$. In Appendix A.4, we even derive high-probability guarantees on this convergence.

### 3.3. Scalable Power Sampling

Based on the above results, this section develops a practical algorithm for scalable, efficient trajectory sampling from LLM power distributions. Algorithm 1 presents our pseudocode for the single-token setting, while Algorithm 2 in Appendix B extends this implementation to batched token settings. Algorithm 1 approximates the global power distribution via sequential, local estimations. At each autoregressive step $t$, the algorithm identifies a set of promising candidates, $\mathcal{G}_t$, using a Top-$K_t$ filter on the base model's logits. To evaluate the long-term viability of each candidate under the power objective, we perform $M_t$ independent Monte Carlo lookahead rollouts. We can further restrict the

**Algorithm 1** Scalable Power Sampling for a Query $q$

1: **Inputs:** Power distribution exponent $\alpha \in [1, \infty)$, upper bounds on trajectory length $T$, size of groups $\{K_t\}_{t=0}^{T-1}$, and total budgets $\{M_t\}_{t=0}^{T-1}$
2: **for** $t = 0 \to T - 1$ **do**
3:     Compute $\mathcal{G}_t(\boldsymbol{x}_{0:t-1}) = \text{Top@}K_t\,[p(\cdot|\boldsymbol{q}, \boldsymbol{x}_{0:t-1})]$
4:     **for** $x_t' \in \mathcal{G}_t(\boldsymbol{x}_{0:t-1})$ **do**
5:         Sample $M_t$ i.i.d. completions
6:         Compute $\hat{\zeta}_t(x_t')$ and $\hat{\zeta}_{t,-s}^{(\text{LOO})}(x_t')$ from Eq. 3 and 6
7:     **end for**
8:     Set $\mathcal{V} = \mathcal{G}_t(\boldsymbol{x}_{0:t-1})$ in Eq. 5
9:     Compute the Jackknife estimator
10:     Sample $x_t \in \mathcal{G}_t(\boldsymbol{x}_{0:t-1})$ w.p. $\hat{p}_{\alpha,\text{ JK}}^{(\text{pow})}(x_t|\boldsymbol{q}, \boldsymbol{x}_{0:t-1})$
11:     Concatenate the sampled tokens $\boldsymbol{x}_{0:t} = \boldsymbol{x}_{0:t-1} \oplus x_t$
12: **end for**
13: Sample the $T^{\text{th}}$ token using low temperature sampling
14: Concatenate the sampled token $\boldsymbol{x}_{0:T} = \boldsymbol{x}_{0:T-1} \oplus x_T$
15: **Return** Sampled trajectory $\boldsymbol{x}_{0:T} = \{x_0, \ldots, x_T\}$

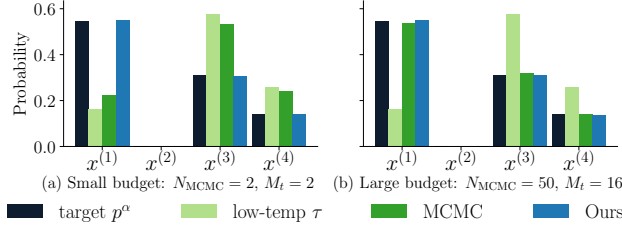

(a) Small budget: $N_{\text{MCMC}} = 2$, $M_t = 2$    (b) Large budget: $N_{\text{MCMC}} = 50$, $M_t = 16$

target $p^\alpha$    low-temp $\tau$    MCMC    Ours

*Figure 1.* Toy example comparing the target power distribution $p^\alpha$, the low-temperature distribution, and the empirical histograms of MCMC and our method ($\alpha = 4$, $\tau = 1/\alpha = 0.25$).

rollout length to the next $H_t$ tokens (see Appendix B). Such truncation further reduces inference times. These trajectories are aggregated to compute likelihood scaling factors, which are subsequently refined using a Jackknife estimator ($\hat{p}_{\alpha,\text{JK}}^{(\text{pow})}$) to mitigate finite-sample bias. The next token $x_t$ is then sampled according to these bias-corrected probabilities, effectively steering the generation toward high-likelihood region token-by-token. Finally, we note that since we evaluate $K_t$ candidates using $M_t$ independent rollouts at each step $t$, the total inference burden is bounded by $\mathcal{O}\left(\sum_{t=0}^{T-1} K_t M_t\right)$ generated sequences. This structured cost offers a distinct efficiency advantage over rejection sampling (Best-of-N), which scales linearly as $\mathcal{O}(N)$ full trajectories. Unlike Best-of-N, which must "pay" for $N$ complete generations even when errors occur early, our formulation allows for targeted compute allocation, investing budget $(K_t, M_t)$ specifically at critical decision points to steer generation, thereby achieving high-fidelity sampling without brute-force costs.

The scalable implementation introduces additional bias relative to the true power distribution through Top-$K_t$ truncation and finite lookahead $H_t$. We further quantify the bias induced by the scalable approximation and discuss the numerical stability of the jackknife estimator in Appendix B.

## 4. Experiments & Results

### 4.1. Building Intuition: Planning vs. Guessing

To build intuition, we consider a simple reasoning problem: "what is the result of 2+2". In this example, the model generates sequences of fixed length $L = 4$ from:

$$\mathcal{V} = \{\texttt{PLAN}, \texttt{GUESS}, \texttt{CALC}, \texttt{ANSWER4}, \texttt{ANSWER5}, \texttt{EOS}\}.$$

We imagine the case when the base model assigns a higher probability to a guessing strategy than to an explicit planning strategy, reflecting a locally plausible but globally suboptimal shortcut: $p(\texttt{PLAN}) = 0.4$, and $p(\texttt{GUESS}) = 0.6$.

If the model selects $\texttt{PLAN}$, it deterministically transitions to $\texttt{CALC}$, i.e., $p(\texttt{CALC} \mid \texttt{PLAN}) = 1$. Conditioned on this prefix, the model produces the correct answer with high probability, such that: $p(\texttt{ANSWER4} \mid \texttt{PLAN}, \texttt{CALC}) = 0.95$, and the incorrect answer ($\texttt{ANSWER5}$) with complementary probability. In contrast, if the model directly selects $\texttt{GUESS}$, it outputs: $p(\texttt{ANSWER4} \mid \texttt{GUESS}) = 0.55$, again followed by the end-of-sequence token $\texttt{EOS}$. All sequences terminate after producing an answer token.

In this toy problem, the model faces an early choice between a locally attractive shortcut and a more structured reasoning strategy. Although $\texttt{GUESS}$ has a higher probability than $\texttt{PLAN}$ at the first step, committing to the $\texttt{PLAN} \to \texttt{CALC}$ trajectory leads to a much higher chance of correct answers.

Concretely, the four feasible sequences are

$$x^{(1)} = (\texttt{PLAN}, \texttt{CALC}, \texttt{ANSWER4}, \texttt{EOS}),$$
$$x^{(2)} = (\texttt{PLAN}, \texttt{CALC}, \texttt{ANSWER5}, \texttt{EOS}),$$
$$x^{(3)} = (\texttt{GUESS}, \texttt{ANSWER4}, \texttt{EOS}, \texttt{EOS}),$$
$$x^{(4)} = (\texttt{GUESS}, \texttt{ANSWER5}, \texttt{EOS}, \texttt{EOS}),$$

with base-model probabilities

$$p(x^{(1)}) = 0.38 \quad p(x^{(2)}) = 0.02$$
$$p(x^{(3)}) = 0.33 \quad p(x^{(4)}) = 0.27.$$

If we apply low-temperature decoding with $\alpha = 4$ (i.e., $\tau = 0.25$), the model becomes confident in its *local* preference and overwhelmingly selects $\texttt{GUESS}$ at the first step:

$$p_\tau(\texttt{PLAN}) = \frac{0.4^4}{0.4^4 + 0.6^4} \approx 0.165, \quad p_\tau(\texttt{GUESS}) \approx 0.835.$$

As a result, low-temperature sampling amplifies the shortcut, despite its lower probability of leading to a correct answer.

In contrast, the power distribution $\pi(x) \propto p(x)^4$ evaluates entire trajectories rather than individual tokens. Under this

distribution, the structured PLAN branch dominates:

$$\pi(x^{(1)}) \approx 0.55, \quad \pi(x^{(3)}) \approx 0.31,$$
$$\pi(x^{(4)}) \approx 0.14, \quad \pi(x^{(2)}) \approx 0.$$

This illustrates why low-temperature decoding can fail to recover the desired power distribution: sharpening local probabilities does not account for downstream consequences. While both MCMC-based power sampling and our estimator-based method converge to $\pi$ with sufficient computation, their finite-budget behaviour differs substantially. MCMC tends to inherit the bias of its low-temperature proposal when few steps are used, whereas our rollout-based future correction explicitly reweights early decisions based on expected future outcomes, yielding samples that more closely match $\pi$ even at small budgets (see Figure 1).

### 4.2. Large-Scale Evaluation

We evaluate our scalable power-distribution sampling approximation on large-scale benchmarks. We assess whether training- and verifier-free autoregressive sampling can recover RL-based post-training gains without MCMC's inference overhead. Comparing against standard decoding, low-temperature sampling, Best-of-N, and MCMC, we characterise accuracy–latency trade-offs across mathematics, code generation, and knowledge-intensive QA.

**Models.** We test four base models to assess robustness across families and training regimes. We use Qwen2.5-7B (Team et al., 2024) to assess general reasoning, and two math-focused models, Qwen2.5-Math-7B (Team et al., 2024) and DeepSeek-Math-7B (Shao et al., 2024), to examine transferability across fine-tuning pipelines. Finally, we evaluate the RL-tuned DeepSeek-Math-7B-RL (via GRPO) (Shao et al., 2024) to determine if our method remains effective on models already sharpened by RL.

**Benchmarks.** We use three complementary benchmarks. For mathematics, we use MATH500 (Lightman et al., 2023) to assess multi-step problem solving via a final-answer exact match. For code, we employ HumanEval (Chen et al., 2021), comprising 164 Python tasks assessed via functional correctness. For QA, we use GPQA-diamond (Rein et al., 2024), a set of 198 expert-authored questions in biology, physics, and chemistry designed to challenge non-experts.

**Baselines.** For all methods, we cap the maximum generation length at $T_{max} = 3072$ tokens, as suggested in Karan & Du (2025); generation may terminate earlier upon emission of the EOS token. Full hyperparameter settings are provided in Appendix C. For low-temperature decoding (Wang et al., 2020), we set the sampling temperature to $1/\alpha = 0.25$ and sample autoregressively from $p_\alpha^{(\text{low. temp})}$. For the Best-

| | MATH500 | HumanEval | GPQA |
|---|---|---|---|
| **Qwen2.5-7B** | | | |
| Base | 0.498 | 0.329 | 0.278 |
| Low-temperature | 0.628 | 0.524 | 0.303 |
| Best-of-N | 0.650 | 0.609 | 0.282 |
| Self-Consistency | 0.670 | 0.742 | 0.282 |
| MCMC Power Sampling | 0.706 | 0.622 | 0.318 |
| **Power Sampling (ours)** | **0.708** | **0.756** | **0.349** |
| GRPO (MATH) | 0.740 | 0.561 | 0.354 |
| **Qwen2.5-Math-7B** | | | |
| Base | 0.496 | 0.329 | 0.278 |
| Low-temperature | 0.690 | 0.512 | 0.353 |
| Best-of-N | 0.684 | 0.512 | 0.343 |
| Self-Consistency | **0.789** | 0.591 | 0.384 |
| MCMC Power Sampling | 0.748 | 0.573 | 0.389 |
| **Power Sampling (ours)** | 0.758 | **0.604** | **0.409** |
| GRPO (MATH) | 0.785 | 0.537 | 0.399 |
| **DeepSeek-Math-7B** | | | |
| Base | 0.362 | 0.415 | 0.333 |
| Low-temperature | 0.366 | 0.427 | **0.430** |
| Best-of-N | 0.420 | 0.433 | 0.338 |
| Self-Consistency | 0.462 | 0.470 | 0.333 |
| MCMC Power Sampling | 0.424 | 0.470 | 0.345 |
| **Power Sampling (ours)** | **0.464** | **0.487** | 0.364 |
| GRPO (MATH) | 0.492 | 0.524 | 0.333 |

*Table 1.* Performance comparison of Power Sampling against base models and sampling baselines.

of-N baseline (Stiennon et al., 2020), we generate 32 independent completions and select the candidate with the highest model score under the base model's log-likelihood. For Self-Consistency (Wang et al., 2022), we generate 16 independent completions and select the final answer by majority voting over extracted answers. For MCMC power sampling, we follow the implementation and hyperparameters of Karan & Du (2025): $\alpha = 4$, block size $B = 192$ (i.e., $T_{max}/16$), $N_{MCMC} = 10$ MCMC steps per stage.

Our method uses the same settings as MCMC power sampling for a fair comparison ($\alpha = 4$, $B = 192$) and performs autoregressive power sampling with block-level lookahead (see Appendix B). At each step, we restrict attention to the Top-$K_t = 8$ candidate tokens, estimate future-scaling factors using $M_t = 8$ Monte Carlo rollouts per candidate.

For RL baselines, we use the GRPO-trained checkpoint from Shao et al. (2025) for the Qwen and the publicly released DeepSeek-Math-7B-RL model for DeepSeek, both of which are post-trained on the MATH training split. We report these post-trained models as GRPO(MATH) in Table 1.

**Main Results.** Table 1 summarises pass@1 accuracies on MATH500, HumanEval, and GPQA across the four models. Relative to standard decoding, low-temperature sampling generally improves performance, but the gains are modest in some settings, suggesting that purely local token-level sharpening has limited ability to extract higher-quality trajectories across tasks. The Best-of-N baseline improves over low-temperature decoding in several regimes, but remains con-

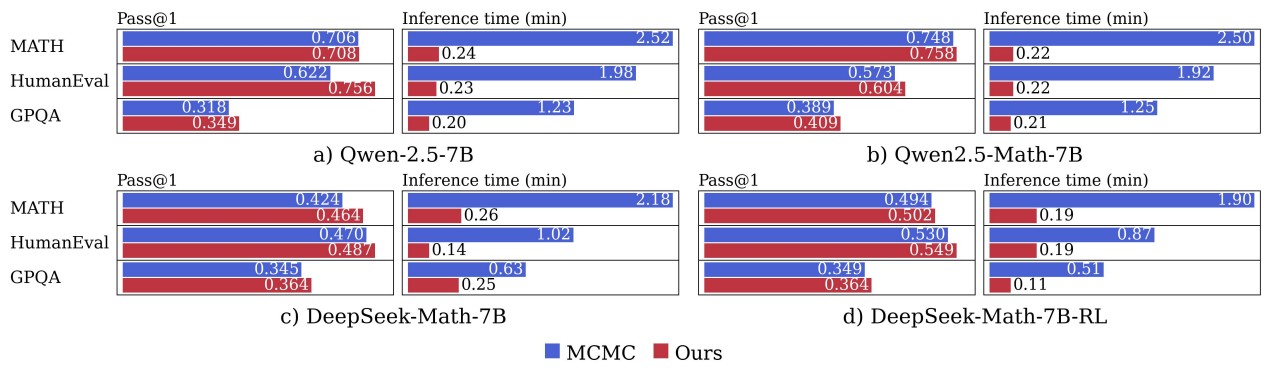

*Figure 2.* Pass@1 performance and per-prompt inference time for Qwen2.5-7B (top left), Qwen2.5-Math-7B (top right), DeepSeek-Math-7B (bottom left), and DeepSeek-Math-7B-RL (bottom right).

sistently weaker than power-sampling-based methods. This behaviour is expected, as selecting the highest-likelihood completion under the base model does not explicitly target the globally sharpened power distribution. Self-Consistency is a strong training-free and reward-free baseline and even outperforms GRPO on MATH500 with Qwen2.5-Math-7B, suggesting that answer-level voting is effective when diverse samples contain multiple correct solution paths. However, it still underperforms power sampling overall, as it aggregates final answers rather than modifying the trajectory-level sampling distribution. Power sampling methods show the best performance across all the sampling methods. Although MCMC and our method target the same power distribution in the limit of infinite compute, their finite-budget behaviour differs, consistent with the intuition in Section 4.1. With the same sampling budget, our estimator more effectively concentrates probability mass on high-quality trajectories, matching or exceeding MCMC performance with up to +13.4%.

Following the evaluation protocol of Karan & Du (2025), we further examine the in-domain and out-of-domain behaviour of reinforcement-learning-based post-training; rows titled GRPO(MATH) in Table 1. Specifically, we consider a GRPO model trained exclusively on mathematical data and evaluate its generalisation performance not only on MATH500, but also on the out-of-domain tasks of code generation and knowledge-intensive question answering. This setting allows us to assess whether improvements induced by reward-based post-training transfer extend beyond the domain on which the reward signal was defined. Overall, our results reproduce the conclusions of Karan & Du (2025) regarding the effectiveness of MCMC-based power sampling, suggesting that a substantial fraction of the gains attributed to reward-based post-training can be recovered through principled sampling from the base model alone.

**Power Sampling Post-Trained LLMs.** Table 2 examines whether power sampling can further improve performance

when applied to an already post-trained model. Specifically, we consider DeepSeek-Math-7B-RL (Shao et al., 2024), which has been post-trained via GRPO on the MATH training split, and evaluate whether power-based sampling remains effective beyond standard decoding.

| | MATH500 | HumanEval | GPQA |
|---|---|---|---|
| **DeepSeek-Math-7B-RL** | | | |
| Base | 0.492 | 0.524 | 0.333 |
| Low-temperature | 0.412 | 0.524 | 0.303 |
| Best-of-N | 0.492 | 0.507 | 0.297 |
| Self-Consistency | 0.492 | 0.524 | 0.349 |
| MCMC Power Sampling | 0.494 | 0.530 | 0.349 |
| **Power Sampling (ours)** | **0.502** | **0.549** | **0.364** |

*Table 2.* Pass@1 accuracy on MATH500, HumanEval, and GPQA for DeepSeek-Math-7B-RL (GRPO-MATH).

Across all three benchmarks, power sampling on top of the GRPO-trained model yields gains over standard decoding. Notably, low-temperature sampling degrades performance on MATH500 and GPQA (e.g., MATH500: $0.492 \rightarrow 0.412$), Best-of-N and Self-Consistency also yield only limited gains on this post-trained model, suggesting that GRPO post-training has already sharpened the token distribution and reduced answer diversity, making local temperature scaling and answer-level voting less effective. In contrast, both MCMC power sampling and our autoregressive approximation provide small improvements, with our method achieving the best performance across all benchmarks. Overall, these results indicate that power-based sampling can still enhance an RL-post-trained model, though the magnitude of improvement is smaller than for non-post-trained LLMs, consistent with the view that GRPO already performs a form of distribution sharpening.

**Inference Efficiency.** Having established that we achieve improved pass@1 performance, we now turn to understanding the inference-time efficiency gains of our approximation. Figure 2 reports inference time for MCMC and our autoregressive approximation across all four models and three

| | MATH500 | HumanEval | GPQA |
|---|---|---|---|
| **Qwen2.5-32B** | | | |
| Base | 0.577 | 0.392 | 0.376 |
| Low-temperature | 0.682 | 0.507 | 0.414 |
| MCMC Power Sampling | 0.738 | 0.567 | 0.383 |
| **Power Sampling (ours)** | **0.748** | **0.793** | **0.429** |

*Table 3.* Pass@1 accuracy on MATH500, HumanEval, and GPQA with Qwen2.5-32B.

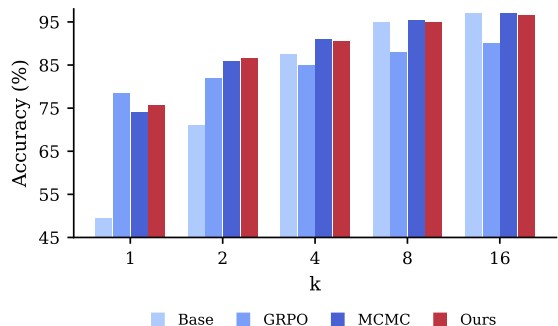

*Figure 3.* Pass@K performance on MATH500 with Qwen2.5-Math-7B of the base model, GRPO, MCMC, and our method. GRPO improves pass@1 but quickly shows diversity collapse.

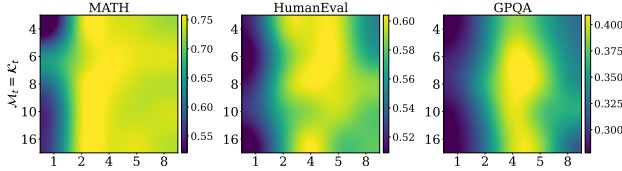

*Figure 4.* Pass@1 on MATH500, HumanEval, and GPQA for Qwen2.5-Math-7B under different hyperparameters.

benchmarks. Across all settings, our method is consistently faster, achieving speedups of up to approximately 10×. In the slowest MCMC regime (Qwen2.5-Math-7B on MATH500), MCMC requires an average of 2.5 minutes per prompt, compared to 0.22 minutes for our method. Even in the closest regime (DeepSeek-Math-7B on GPQA), MCMC still incurs a substantially higher cost, taking 0.63 minutes per prompt versus 0.25 minutes for our approach. We also find that the final completion lengths are similar across methods: in a representative MATH500 setting with Qwen2.5-Math-7B, base decoding generates 640 tokens on average, compared with 682 for the GRPO model, and 690 for Self-Consistency, 679 for MCMC and 667 for our method. These results indicate that the observed speedups result from eliminating iterative MCMC refinement rather than differences in output length, while preserving the benefits of power-based sampling.

**Scaling to Larger Models.** We further evaluate our method on Qwen2.5-32B to examine its scalability. As shown in Table 3, our method consistently improves over base and low-temperature decoding across all three benchmarks. Compared with MCMC Power Sampling, it achieves a 9.4% average improvement while remaining substantially faster, requiring 0.36 minutes per prompt versus 4.1 minutes for MCMC. These results suggest that our autoregressive approximation remains effective at larger model scale.

### 4.3. Diversity Analysis & Ablation

**Pass@K Results.** We report pass@K performance for K∈ {1, 2, 4, 8, 16} by sampling K independent completions

per prompt and marking a problem as solved if any of the completions is correct. We summarise the results on the MATH500 obtained with Qwen2.5-Math-7B in Figure 3. Further Pass@K results can be found in Appendix C.

We note that although GRPO (Shao et al., 2025) yields a strong boost at pass@1, its gains diminish as K increases, consistent with recent observations of *diversity collapse* (mode-seeking behaviour that reduces coverage of alternative solution paths and suppresses pass@K performance (Yue et al., 2025)). In contrast, power-distribution sampling sharpens the sampling distribution at inference time without updating parameters, improving pass@1 over the base model while better preserving diversity; our method closely tracks base model performance for larger K (e.g., K≥ 8) and is stronger at small K (notably K∈ {1, 2, 4}), matching patterns reported in prior power-sampling work (Karan & Du, 2025). We further quantify this effect by measuring answer repetition across 8 rollouts per prompt on 100 MATH500 examples with Qwen2.5-Math-7B: the duplicate-answer rate is 58.3% for low-temperature decoding, 62.5% for our method, and 83.3% for GRPO, indicating that our inference-time sharpening preserves substantially more answer diversity than RL post-training.

**Ablation Study.** We analyse sensitivity to the power exponent $\alpha$, rollout budget $M_t$, and candidate set size $K_t$. Figure 4 shows the method is robust to $K_t$ and $M_t$, where moderate values offer a favourable accuracy–compute trade-off. In contrast, performance is sensitive to $\alpha$. Intermediate values (e.g., $\alpha \in \{4, 5\}$) yield the strongest results, consistent with MCMC findings (Karan & Du, 2025), surpassing both the base distribution ($\alpha = 1$) and overly aggressive sharpening ($\alpha = 8$). We therefore fix $\alpha = 4$ and $K_t = M_t = 8$ in our experiments. Further analysis, results on additional models, and ablations on the truncated lookahead horizon $H_t$ and block size $B$ are provided in Appendix C.

We also ablate the jackknife estimator to evaluate its importance. Across three base models and three benchmarks, replacing the jackknife estimator with the naive rollout estimator leads to an average drop of 4.29%, confirming that bias correction is an important component of the proposed approximation. Detailed results are provided in Appendix C.

# 5. Related Works

**Reinforcement Learning and Distribution Sharpening.**
RL post-training has emerged as a dominant paradigm for enhancing the reasoning capabilities of LLMs. Approaches such as Rejection Sampling Fine-tuning (Liu et al., 2023; Xiong et al., 2025) and GRPO and its variants (Shao et al., 2024; Yu et al., 2025) optimise models against automated verifiers; see the surveys in Xu et al. (2025); Zhang et al. (2025) for potential applications and an in-depth discussion.

**Inference-Time Reasoning and Sampling.** Various inference-time strategies exist to bypass training complexity. While low-temperature sampling (Wang et al., 2020) sharpens local probabilities, it ignores global quality, and Best-of-N (Brown et al., 2024; Huang et al., 2025) only implicitly targets rewards; we outperform both (see Section 4.2). Although our use of Monte Carlo rollouts shares conceptual roots with Tree of Thoughts (Yao et al., 2023) and MCTS (Zhang et al., 2023; 2024; Cao et al., 2025), our approach avoids their complex state tracking and backtracking, remaining a pure *sampling method* without persistent memory. Similarly, while speculative (Leviathan et al., 2023; Chen et al., 2023; Zimmer et al., 2024) and lookahead decoding (Fu et al., 2024) use "draft-then-verify" parallelism like our batched algorithm (Appendix B), they aim to preserve $p(x)$ for speed, whereas we explicitly alter it to $p^\alpha(x)$ for reasoning. Finally, inference-aware training (Vijayakumar et al., 2016; Wang et al., 2022; Chow et al., 2024) is orthogonal to our method and can be combined in the future.

Most relevantly, Karan & Du (2025) utilised the concept of *Power Distribution Sampling*, proving that sampling from a globally sharpened distribution $p^\alpha$ allows base models to match or exceed the performance of RL-trained models. However, their proposed implementation relies on MCMC, which suffers from high latency due to iterative resampling and slow mixing times in high-dimensional token spaces (Levin & Peres, 2017; Gheissari et al., 2018). Our approach targets the same theoretical distribution but employs a novel autoregressive approximation, eliminating the need for iterative MCMC and significantly reducing inference latency. Theoretically, targeting $p^\alpha$ aligns with sampling from an annealed or tempered distribution (Neal, 2001), a concept widely used in statistical physics and diffusion models to avoid mode collapse. While recent works have applied annealing to diffusion guidance, we demonstrate its efficacy for discrete autoregressive reasoning. We also note that sampling from the power distribution can also be viewed as sampling from an energy-based model with inverse temperatures; see Ackley et al. (1985); LeCun et al. (2006).

# 6. Conclusions and Future Work

We demonstrate that latent reasoning capabilities in base models are accessible via *distribution sharpening*. By bridging global distributions and local policies (Theorem 3.1), our training-free sampler matches GRPO on MATH500, HumanEval, and GPQA. It achieves this with over $10\times$ lower latency than MCMC, providing a scalable alternative to post-training pipelines.

Future work includes reducing variance via control variates (Lavenberg & Welch, 1981), optimising compute via adaptive budgets (Alomrani et al., 2025), and amortising rollout costs with speculative decoding (Leviathan et al., 2023). We also plan to extend our method to agentic settings (Lu et al., 2024; Grosnit et al., 2025; Yuksekgonul et al., 2026).

# Impact Statement

**Democratisation of Advanced Reasoning.** The primary impact of this work is the democratisation of high-performance reasoning capabilities. By demonstrating that base models can match the performance of RL-finetuned models (e.g., GRPO) through inference-time sampling alone, we reduce the necessity for massive computational resources typically required for post-training alignment. This lowers the barrier to entry for academic researchers and smaller organisations, allowing them to deploy state-of-the-art reasoning systems without access to large-scale GPU clusters for training.

**Environmental Considerations.** From an environmental perspective, our approach offers a favourable trade-off. While our autoregressive lookahead mechanism increases inference cost relative to standard decoding, it eliminates the substantial carbon footprint associated with iterative RL fine-tuning and reduces inference energy consumption by an order of magnitude compared to MCMC-based baselines. We believe that this represents a significant step towards "Green AI" for high-reliability tasks.

**Safety and Dual-Use Risks.** However, our method also introduces specific safety considerations. Because our algorithm amplifies the latent probability mass of the base model ("distribution sharpening") without an external reward model or human feedback loop, it relies entirely on the safety alignment of the underlying base model. If a base model contains latent hazardous knowledge or biases, our method could theoretically "sharpen" these behaviours just as effectively as it sharpens mathematical reasoning. Consequently, we strongly recommend that this sampling strategy be applied to base models that have undergone rigorous safety filtering or instruction tuning before deployment.

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

# A. Theoretical Results

## A.1. Proof of Theorem 3.1

**Theorem A.1.** *Let $p(x)$ be a pretrained large language model, $\boldsymbol{q}$ an input prompt and $\alpha > 1$ be an exponent parameter. Then, for any partially generated sequence $\boldsymbol{x}_{0:t-1} = \{x_0, \ldots, x_{t-1}\}$ we have:*

$$p_\alpha^{(pow)}(x_t|\boldsymbol{q}, \boldsymbol{x}_{0:t-1}) = \frac{p^\alpha(x_t|\boldsymbol{q}, \boldsymbol{x}_{0:t-1})\zeta_t(x_t)}{\sum_{x_t' \in \mathcal{V}} p^\alpha(x_t'|\boldsymbol{q}, \boldsymbol{x}_{0:t-1})\zeta_t(x_t')}. \tag{7}$$

*where for each token $x_t' \in \mathcal{V}$, we have that: $\zeta_t(x_t') = \sum_{\boldsymbol{x}_{t+1:T}} p^\alpha(\boldsymbol{x}_{t+1:T}|\boldsymbol{q}, \boldsymbol{x}_{0:t-1}, x_t')$.*

*Proof.* Let us fix some $t \in [0, T)$ and consider the partially generated sequence $\boldsymbol{x}_{0:t-1}$ such that:

$$\boldsymbol{x}_{0:t-1} = \begin{cases} x_0, \ldots, x_{t-1}, & \text{if } t \geq 1, \\ \{\}, & \text{if } t = 0. \end{cases}$$

For a fixed token $x_t$, let us use the following notation:

$$\boldsymbol{x}_{0:t} = \{x_0, \ldots, x_{t-1}, x_t\}, \quad \boldsymbol{x}_{0:t-1}, x_t' = \{x_0, \ldots, x_{t-1}, x_t'\}, \quad \boldsymbol{x}_{t:T} = \{x_t, \ldots, x_T\}.$$

Consider the conditional probability:

$$
\begin{aligned}
p_\alpha^{(\text{pow})}(x_t|\boldsymbol{q}, \boldsymbol{x}_{0:t-1}) &= \frac{\sum_{\boldsymbol{x}_{t+1:T}} p^\alpha(\boldsymbol{q}, \boldsymbol{x}_{0:t-1}, x_t, \boldsymbol{x}_{t+1:T})}{\sum_{\boldsymbol{x}_{t:T}} p^\alpha(\boldsymbol{q}, \boldsymbol{x}_{0:t-1}, \boldsymbol{x}_{t:T})} \\
&= \frac{\sum_{\boldsymbol{x}_{t+1:T}} p^\alpha(\boldsymbol{q}, \boldsymbol{x}_{0:t}, \boldsymbol{x}_{t+1:T})}{\sum_{\boldsymbol{x}_{t+1:T}} p^\alpha(\boldsymbol{q}, \boldsymbol{x}_{0:t}, \boldsymbol{x}_{t+1:T}) + \sum_{x_t' \neq x_t} \sum_{\boldsymbol{x}_{t+1:T}} p^\alpha(\boldsymbol{q}, \boldsymbol{x}_{0:t-1}x_t', \boldsymbol{x}_{t+1:T})} \\
&= \frac{1}{1 + \sum_{x_t' \neq x_t} \frac{\sum_{\boldsymbol{x}_{t+1:T}} p^\alpha(\boldsymbol{q}, \boldsymbol{x}_{0:t-1}x_t', \boldsymbol{x}_{t+1:T})}{\sum_{\boldsymbol{x}_{t+1:T}} p^\alpha(\boldsymbol{q}, \boldsymbol{x}_{0:t}, \boldsymbol{x}_{t+1:T})}} \\
&= \frac{1}{1 + \sum_{x_t' \neq x_t} \frac{\sum_{\boldsymbol{x}_{t+1:T}} p^\alpha(\boldsymbol{x}_{t+1:T}|\boldsymbol{q}, \boldsymbol{x}_{0:t-1}x_t')p^\alpha(x_t'|\boldsymbol{q}, \boldsymbol{x}_{0:t})p^\alpha(\boldsymbol{q}, \boldsymbol{x}_{0:t-1})}{\sum_{\boldsymbol{x}_{t+1:T}} p^\alpha(\boldsymbol{x}_{t+1:T}|\boldsymbol{q}, \boldsymbol{x}_{0:t})p^\alpha(x_t|\boldsymbol{q}, \boldsymbol{x}_{0:t-1})p^\alpha(\boldsymbol{q}, \boldsymbol{x}_{0:t-1})}} \\
&= \frac{1}{1 + \sum_{x_t' \neq x_t} \frac{p^\alpha(x_t'|\boldsymbol{q}, \boldsymbol{x}_{0:t-1})}{p^\alpha(x_t|\boldsymbol{q}, \boldsymbol{x}_{0:t-1})} \frac{\sum_{\boldsymbol{x}_{t+1:T}} p^\alpha(\boldsymbol{x}_{t+1:T}|\boldsymbol{q}, \boldsymbol{x}_{0:t-1}x_t')}{\sum_{\boldsymbol{x}_{t+1:T}} p^\alpha(\boldsymbol{x}_{t+1:T}|\boldsymbol{q}, \boldsymbol{x}_{0:t})}} = \frac{1}{1 + \sum_{x_t' \neq x_t} \frac{p^\alpha(x_t'|\boldsymbol{q}, \boldsymbol{x}_{0:t-1})}{p^\alpha(x_t|\boldsymbol{q}, \boldsymbol{x}_{0:t-1})} \frac{\zeta_t(x_t')}{\zeta_t(x_t)}} \\
&= \frac{p^\alpha(x_t|\boldsymbol{q}, \boldsymbol{x}_{0:t-1})\zeta_t(x_t)}{\sum_{x_t' \in \mathcal{V}} p^\alpha(x_t'|\boldsymbol{q}, \boldsymbol{x}_{0:t-1})\zeta_t(x_t')}.
\end{aligned}
$$

To complete the proof, let us consider the last token case. For the sequence of tokens $\boldsymbol{x} = x_1, \ldots, x_T$ we have:

$$
\begin{aligned}
p_\alpha^{(\text{pow})}(x_T|\boldsymbol{q}, x_0, \ldots x_{T-1}) &= \frac{p^\alpha(\boldsymbol{q}, \boldsymbol{x}_{0:T-1}, x_T)}{\sum_{x_T' \in \mathcal{V}} p^\alpha(\boldsymbol{q}, \boldsymbol{x}_{0:T-1}, x_T')} \\
&= \frac{p^\alpha(x_T|\boldsymbol{q}, \boldsymbol{x}_{0:T-1})p^\alpha(\boldsymbol{q}, \boldsymbol{x}_{0:T-1})}{\sum_{x_T' \in \mathcal{V}} p^\alpha(x_T'|\boldsymbol{q}, \boldsymbol{x}_{0:T-1})p^\alpha(\boldsymbol{q}, \boldsymbol{x}_{0:T-1})} = \frac{p^\alpha(x_T|\boldsymbol{q}, x_0, \ldots, x_{T-1})}{\sum_{x_T' \in \mathcal{V}} p^\alpha(x_T'|\boldsymbol{q}, x_0, \ldots, x_{T-1})} \\
&= p_\alpha^{(\text{low. temp})}(x_T|\boldsymbol{q}, \boldsymbol{x}_{0:T-1}).
\end{aligned}
$$

$\square$

## A.2. Proof of Lemma 3.2

**Lemma A.2.** *Let $\hat{p}_\alpha^{(pow)}(x_t|\boldsymbol{q}, \boldsymbol{x}_{0:t-1})$ be the Monte Carlo estimator of the power distribution defined in Equation 4. Then:*

$$\mathbb{E}\left[\hat{p}_\alpha^{(pow)}(x_t|\boldsymbol{q}, \boldsymbol{x}_{0:t-1})\right] = p_\alpha^{(pow)}(x_t|\boldsymbol{q}, \boldsymbol{x}_{0:t-1}) + \mathcal{O}\left(\frac{1}{M_t}\right).$$

*Proof.* First, we note that:

$$p_\alpha^{(\text{pow})}(x_t|\boldsymbol{q}, \boldsymbol{x}_{0:t-1}) = \frac{\mathbb{E}\left[p^\alpha(x_t|\boldsymbol{q}, \boldsymbol{x}_{0:t-1})\sum_{r=1}^{M_t} p^{\alpha-1}(\boldsymbol{x}_{t+1:T}^{[r]}|\boldsymbol{q}, \boldsymbol{x}_{0:t-1}, x_t)\right]}{\mathbb{E}\left[\sum_{x_t' \in \mathcal{V}} p^\alpha(x_t'|\boldsymbol{q}, \boldsymbol{x}_{0:t-1})\sum_{r=1}^{M_t} p^{\alpha-1}(\boldsymbol{x}_{t+1:T}^{[r]}|\boldsymbol{q}, \boldsymbol{x}_{0:t-1}, x_t')\right]} = \frac{\zeta_t(x_t)p^\alpha(x_t|\boldsymbol{q}, \boldsymbol{x}_{0:t-1})}{\sum_{x_t' \in \mathcal{V}} \zeta_t(x_t')p^\alpha(x_t'|\boldsymbol{q}, \boldsymbol{x}_{0:t-1})}.$$

Let us denote $\delta(x_t) = \frac{1}{M_t}\sum_{r=1}^{M_t} p^\alpha(x_t|\boldsymbol{q}, \boldsymbol{x}_{0:t-1})p^{\alpha-1}(\boldsymbol{x}_{t+1:T}^{[r]}|\boldsymbol{q}, \boldsymbol{x}_{0:t-1}, x_t) - \zeta_t(x_t)p^\alpha(x_t|\boldsymbol{q}, \boldsymbol{x}_{0:t-1})$. Then, we have:

$$\hat{p}_\alpha^{(\text{pow})}(x_t|\boldsymbol{q}, \boldsymbol{x}_{0:t-1}) = \frac{\zeta_t(x_t)p^\alpha(x_t|\boldsymbol{q}, \boldsymbol{x}_{0:t-1}) + \delta(x_t)}{\sum_{x_t' \in \mathcal{V}}\left[\zeta_t(x_t')p^\alpha(x_t'|\boldsymbol{q}, \boldsymbol{x}_{0:t-1}) + \delta(x_t')\right]}$$

$$= (\zeta_t(x_t)p^\alpha(x_t|\boldsymbol{q}, \boldsymbol{x}_{0:t-1}) + \delta(x_t))\left[\frac{1}{\sum_{x_t' \in \mathcal{V}}\left[\zeta_t(x_t')p^\alpha(x_t'|\boldsymbol{q}, \boldsymbol{x}_{0:t-1}) + \delta(x_t')\right]}\right]$$

$$= \frac{(\zeta_t(x_t)p^\alpha(x_t|\boldsymbol{q}, \boldsymbol{x}_{0:t-1}) + \delta(x_t))}{\left[\sum_{x_t' \in \mathcal{V}} \zeta_t(x_t')p^\alpha(x_t'|\boldsymbol{q}, \boldsymbol{x}_{0:t-1})\right]\left[1 + \frac{\sum_{x_t' \in \mathcal{V}}\delta(x_t')}{\sum_{x_t' \in \mathcal{V}}\zeta_t(x_t')p^\alpha(x_t'|\boldsymbol{q}, \boldsymbol{x}_{0:t-1})}\right]}$$

$$= \frac{\zeta_t(x_t)p^\alpha(x_t|\boldsymbol{q}, \boldsymbol{x}_{0:t-1}) + \delta(x_t)}{\sum_{x_t' \in \mathcal{V}} \zeta_t(x_t')p^\alpha(x_t'|\boldsymbol{q}, \boldsymbol{x}_{0:t-1})}\left[1 + \frac{\sum_{x_t' \in \mathcal{V}}\delta(x_t')}{\sum_{x_t' \in \mathcal{V}}\zeta_t(x_t')p^\alpha(x_t'|\boldsymbol{q}, \boldsymbol{x}_{0:t-1})}\right]^{-1}$$

$$= \frac{\zeta_t(x_t)p^\alpha(x_t|\boldsymbol{q}, \boldsymbol{x}_{0:t-1}) + \delta(x_t)}{\sum_{x_t' \in \mathcal{V}} \zeta_t(x_t')p^\alpha(x_t'|\boldsymbol{q}, \boldsymbol{x}_{0:t-1})} \times$$

$$\times \left[1 - \frac{\sum_{x_t' \in \mathcal{V}}\delta(x_t')}{\sum_{x_t' \in \mathcal{V}}\zeta_t(x_t')p^\alpha(x_t'|\boldsymbol{q}, \boldsymbol{x}_{0:t-1})} + \left[\frac{\sum_{x_t' \in \mathcal{V}}\delta(x_t')}{\sum_{x_t' \in \mathcal{V}}\zeta_t(x_t')p^\alpha(x_t'|\boldsymbol{q}, \boldsymbol{x}_{0:t-1})}\right]^2\right]$$

$$+ \mathcal{O}\left(\left[\sum_{x_t' \in \mathcal{V}}\delta(x_t')\right]^3\right),$$

where in the last step we used the Taylor expansion for $(1 + z)^{-1}$. The above expression can be rewritten as:

$$\hat{p}_\alpha^{(\text{pow})}(x_t|\boldsymbol{q}, \boldsymbol{x}_{0:t-1}) = \frac{\zeta_t(x_t)p^\alpha(x_t|\boldsymbol{q}, \boldsymbol{x}_{0:t-1})}{\sum_{x_t' \in \mathcal{V}} \zeta_t(x_t')p^\alpha(x_t'|\boldsymbol{q}, \boldsymbol{x}_{0:t-1})} + \frac{\delta(x_t)}{\sum_{x_t' \in \mathcal{V}} \zeta_t(x_t')p^\alpha(x_t'|\boldsymbol{q}, \boldsymbol{x}_{0:t-1})} \tag{8}$$

$$- \frac{\zeta_t(x_t)p^\alpha(x_t|\boldsymbol{q}, \boldsymbol{x}_{0:t-1})\sum_{x_t' \in \mathcal{V}}\delta(x_t')}{\left[\sum_{x_t' \in \mathcal{V}} \zeta_t(x_t')p^\alpha(x_t'|\boldsymbol{q}, \boldsymbol{x}_{0:t-1})\right]^2} - \frac{\delta(x_t)\sum_{x_t' \in \mathcal{V}}\delta(x_t')}{\left[\sum_{x_t' \in \mathcal{V}} \zeta_t(x_t')p^\alpha(x_t'|\boldsymbol{q}, \boldsymbol{x}_{0:t-1})\right]^2}$$

$$+ \frac{\zeta_t(x_t)p^\alpha(x_t|\boldsymbol{q}, \boldsymbol{x}_{0:t-1})\left[\sum_{x_t' \in \mathcal{V}}\delta(x_t')\right]^2}{\left[\sum_{x_t' \in \mathcal{V}} \zeta_t(x_t')p^\alpha(x_t'|\boldsymbol{q}, \boldsymbol{x}_{0:t-1})\right]^3} + \mathcal{O}\left(\left[\sum_{x_t' \in \mathcal{V}}\delta(x_t')\right]^3\right).$$

Now, notice:

$$\mathbb{E}[\delta(x_t')] = 0 \quad \text{for all } x_t' \in \mathcal{V},$$

$$\mathbb{E}\left[\delta(x_t)\sum_{x_t' \in \mathcal{V}}\delta(x_t')\right] = \mathbb{E}\left[\delta^2(x_t)\right] = \mathcal{O}\left(\frac{1}{M_t}\right) \quad \text{for all } x_t \in \mathcal{V},$$

$$\mathbb{E}\left[\left[\sum_{x_t' \in \mathcal{V}}\delta(x_t')\right]^2\right] = \mathbb{E}\left[\sum_{x_t' \in \mathcal{V}}\delta^2(x_t') + \sum_{x_t',x_t'': \, x_t' \neq x"_t}\delta(x_t')\delta(x_t'')\right] = \sum_{x_t' \in \mathcal{V}}\mathbb{E}\left[\delta^2(x_t')\right] = \mathcal{O}\left(\frac{1}{M_t}\right),$$

$$\mathbb{E}\left[\left[\sum_{x_t' \in \mathcal{V}}\delta(x_t')\right]^3\right] = \mathcal{O}\left(\frac{1}{M_t}\right).$$

Taking expectation $\mathbb{E}[\cdot]$ in Equation 11 and using the above facts, we get:

$$\mathbb{E}\left[\hat{p}_\alpha^{(\text{pow})}(x_t|\boldsymbol{q}, \boldsymbol{x}_{0:t-1})\right] = \frac{\zeta_t(x_t)p^\alpha(x_t|\boldsymbol{q}, \boldsymbol{x}_{0:t-1})}{\sum_{x_t' \in \mathcal{V}}\zeta_t(x_t')p^\alpha(x_t'|\boldsymbol{q}, \boldsymbol{x}_{0:t-1})} + \mathcal{O}\left(\frac{1}{M_t}\right).$$

Upon using $p_\alpha^{(\text{pow})}(x_t|\boldsymbol{q}, \boldsymbol{x}_{0:t-1}) = \frac{\zeta_t(x_t)p^\alpha(x_t|\boldsymbol{q},\boldsymbol{x}_{0:t-1})}{\sum_{x_t' \in \mathcal{V}}\zeta_t(x_t')p^\alpha(x_t'|\boldsymbol{q},\boldsymbol{x}_{0:t-1})}$ in the above result, we eventually get:

$$\mathbb{E}\left[\hat{p}_\alpha^{(\text{pow})}(x_t|\boldsymbol{q}, \boldsymbol{x}_{0:t-1})\right] = p_\alpha^{(\text{pow})}(x_t|\boldsymbol{q}, \boldsymbol{x}_{0:t-1}) + \mathcal{O}\left(\frac{1}{M_t}\right),$$

which establishes the statement of the lemma. $\qquad\qquad\square$

### A.3. Proof of Lemma 3.3

**Lemma A.3.** *Let* $\hat{p}_{\alpha,\,JK}^{(pow)}(\cdot|\boldsymbol{q}, \boldsymbol{x}_{0:t-1})$ *be the jackknife estimator of the power distribution defined in Equation 5. Then:*

$$\mathbb{E}\left[\hat{p}_{\alpha,\,JK}^{(pow)}(\cdot|\boldsymbol{q}, \boldsymbol{x}_{0:t-1})\right] = p_\alpha^{(pow)}(\cdot|\boldsymbol{q}, \boldsymbol{x}_{0:t-1}) + \mathcal{O}\left(\frac{1}{M_t^2}\right).$$

*Proof.* Let us denote for brevity $\alpha^{[r]}(x_t') = p^\alpha(x_t'|\boldsymbol{q}, \boldsymbol{x}_{0:t-1})p^{\alpha-1}(\boldsymbol{x}_{t+1:T}^{[r]}|\boldsymbol{q}, \boldsymbol{x}_{0:t-1}, x_t')$ for any $x_t' \in \mathcal{V}$. From the previous lemma we can write the following result for the Monte-Carlo estimator:

$$\mathbb{E}\left[\hat{p}_\alpha^{(\text{pow})}(x_t|\boldsymbol{q}, \boldsymbol{x}_{0:t-1})\right] = \mathbb{E}\left[\frac{\frac{1}{M_t}\sum_{r=1}^{M_t}\alpha^{[r]}(x_t)}{\frac{1}{M_t}\sum_{r=1}^{M_t}\sum_{x_t' \in \mathcal{V}}\alpha^{[r]}(x_t')}\right]$$

$$= p_\alpha^{(\text{pow})}(x_t|\boldsymbol{q}, \boldsymbol{x}_{0:t-1}) + \sum_{i=1}^{\infty}\frac{b_i}{M_t^i},$$

for some collection of real numbers $\{b_i\}_{i=1}^{\infty}$. Then, for the jackknife estimator, we can write:

$$\mathbb{E}\left[\hat{p}_{\alpha,\,\text{JK}}^{(\text{pow})}(x_t|\boldsymbol{q},\boldsymbol{x}_{0:t-1})\right] = \mathbb{E}\left[M_t \frac{\frac{1}{M_t}\sum_{r=1}^{M_t}\alpha^{[r]}(x_t)}{\frac{1}{M_t}\sum_{r=1}^{M_t}\sum_{x_t'\in\mathcal{V}}\alpha^{[r]}(x_t')} - \frac{M_t-1}{M_t}\sum_{s=1}^{M_t}\frac{\frac{1}{M_t-1}\sum_{r=1,r\neq s}^{M_t}\alpha^{[r]}(x_t)}{\frac{1}{M_t-1}\sum_{r=1,r\neq s}^{M_t}\sum_{x_t'\in\mathcal{V}}\alpha^{[r]}(x_t')}\right]$$

$$= M_t\mathbb{E}\left[\frac{\frac{1}{M_t}\sum_{r=1}^{M_t}\alpha^{[r]}(x_t)}{\frac{1}{M_t}\sum_{r=1}^{M_t}\sum_{x_t'\in\mathcal{V}}\alpha^{[r]}(x_t')}\right] - \frac{M_t-1}{M_t}\sum_{s=1}^{M_t}\mathbb{E}\left[\frac{\frac{1}{M_t-1}\sum_{r=1,r\neq s}^{M_t}\alpha^{[r]}(x_t)}{\frac{1}{M_t-1}\sum_{r=1,r\neq s}^{M_t}\sum_{x_t'\in\mathcal{V}}\alpha^{[r]}(x_t')}\right]$$

$$= M_t\left[p_{\alpha}^{(\text{pow})}(x_t|\boldsymbol{q},\boldsymbol{x}_{0:t-1}) + \sum_{i=1}^{\infty}\frac{b_i}{M_t^i}\right] - \frac{M_t-1}{M_t}\sum_{s=1}^{M_t}\left[p_{\alpha}^{(\text{pow})}(x_t|\boldsymbol{q},\boldsymbol{x}_{0:t-1}) + \sum_{i=1}^{\infty}\frac{b_i}{(M_t-1)^i}\right]$$

$$= M_t p_{\alpha}^{(\text{pow})}(x_t|\boldsymbol{q},\boldsymbol{x}_{0:t-1}) + \sum_{i=1}^{\infty}\frac{b_i}{M_t^{i-1}} - (M_t-1)p_{\alpha}^{(\text{pow})}(x_t|\boldsymbol{q},\boldsymbol{x}_{0:t-1})$$

$$- (M_t-1)\sum_{i=1}^{\infty}\frac{b_i}{(M_t-1)^i} = p_{\alpha}^{(\text{pow})}(x_t|\boldsymbol{q},\boldsymbol{x}_{0:t-1}) + \sum_{i=1}^{\infty}\left[\frac{b_i}{M_t^{i-1}} - \frac{b_i}{(M_t-1)^{i-1}}\right]$$

$$= p_{\alpha}^{(\text{pow})}(x_t|\boldsymbol{q},\boldsymbol{x}_{0:t-1}) + \sum_{i=2}^{\infty}b_i\frac{(M_t-1)^{i-1}-M_t^{i-1}}{M_t^{i-1}(M_t-1)^{i-1}} = p_{\alpha}^{(\text{pow})}(x_t|\boldsymbol{q},\boldsymbol{x}_{0:t-1}) + \sum_{i=2}^{\infty}b_i c_i.$$

where $c_i = \frac{(M_t-1)^{i-1}-M_t^{i-1}}{M_t^{i-1}(M_t-1)^{i-1}} = \mathcal{O}\left(\frac{1}{M_t^i}\right)$. Hence, we have

$$\mathbb{E}\left[\hat{p}_{\alpha,\,\text{JK}}^{(\text{pow})}(x_t|\boldsymbol{q},\boldsymbol{x}_{0:t-1})\right] = p_{\alpha}^{(\text{pow})}(x_t|\boldsymbol{q},\boldsymbol{x}_{0:t-1}) + \mathcal{O}\left(\frac{1}{M_t^2}\right).$$

$\square$

### A.4. High-Probability Convergence Guarantees

**Corollary A.4.** *Let $\epsilon \in (0,1)$ be any arbitrary precision parameter, $\delta \in (0,1)$ be arbitrary confidence parameter, and let sample complexity $M_t$ for computing the scaling factors $\zeta_t(x_t')$ for any $x_t \in \mathcal{V}$ be equal to $M_t = \mathcal{O}\left(\frac{1}{\epsilon^2}\log\left(\frac{1}{\delta}\right)\right)$. Then, for any fixed token $x_t \in \mathcal{V}$ we have:*

$$\mathbb{P}\left[|\hat{p}_{\alpha,\,\text{JK}}^{(pow)}(x_t|\boldsymbol{q},\boldsymbol{x}_{0:t-1}) - p_{\alpha}^{(pow)}(x_t|\boldsymbol{q},\boldsymbol{x}_{0:t-1})| \leq \epsilon\right] \geq 1 - \delta.$$

*Proof.* Let us introduce the following notation for brevity:

$$\bar{X}_t = \frac{1}{M_t}\sum_{r=1}^{M_t}p^{\alpha-1}(\boldsymbol{x}_{t+1:T}^{[r]}|\boldsymbol{q},\boldsymbol{x}_{0:t-1},x_t), \quad \mathbb{E}[\bar{X}_t] = \mu_t \text{ with } \{\boldsymbol{x}_{t+1:T}^{[r]}\} \sim p(\cdot|\boldsymbol{q},\boldsymbol{x}_{0:t-1},x_t),$$

$$\bar{X} = \frac{1}{M_t}\sum_{r=1}^{M_t}\sum_{x_t'\in\mathcal{V}}p^{\alpha-1}(\boldsymbol{x}_{t+1:T}^{'[r]}|\boldsymbol{q},\boldsymbol{x}_{0:t-1},x_t'), \quad \mathbb{E}[\bar{X}] = \mu \text{ with } \{\boldsymbol{x}_{t+1:T}^{'[r]}\} \sim p(\cdot|\boldsymbol{q},\boldsymbol{x}_{0:t-1},x_t'),$$

$$\bar{X}_{t\setminus s} = \frac{1}{M_t-1}\sum_{r=1,r\neq s}^{M_t}p^{\alpha-1}(\boldsymbol{x}_{t+1:T}^{[r]}|\boldsymbol{q},\boldsymbol{x}_{0:t-1},x_t), \quad \mathbb{E}[\bar{X}_{t\setminus s}] = \mu_t \text{ with } \{\boldsymbol{x}_{t+1:T}^{[r]}\} \sim p(\cdot|\boldsymbol{q},\boldsymbol{x}_{0:t-1},x_t),$$

$$\bar{X}_{\setminus s} = \frac{1}{M_t-1}\sum_{r=1,r\neq s}^{M_t}\sum_{x_t'\in\mathcal{V}}p^{\alpha-1}(\boldsymbol{x}_{t+1:T}^{'[r]}|\boldsymbol{q},\boldsymbol{x}_{0:t-1},x_t'), \quad \mathbb{E}[\bar{X}_{\setminus s}] = \mu \text{ with } \{\boldsymbol{x}_{t+1:T}^{'[r]}\} \sim p(\cdot|\boldsymbol{q},\boldsymbol{x}_{0:t-1},x_t').$$

In these notations, we have

$$\hat{p}_{\alpha,\,\text{JK}}^{(\text{pow})}(x_t|\boldsymbol{q},\boldsymbol{x}_{0:t-1}) = M_t\frac{\bar{X}_t}{\bar{X}} - \frac{M_t-1}{M_t}\sum_{s=1}^{M_t}\frac{\bar{X}_{t\setminus s}}{\bar{X}_{\setminus s}},$$

$$p_{\alpha}^{(\text{pow})}(x_t|\boldsymbol{q},\boldsymbol{x}_{0:t-1}) = \frac{\mu_t}{\mu}.$$

Following a Chernoff bound, for any $\eta \in (0, 1)$ we have:

$$\mathbb{P}\left[|\bar{X}_t - \mu_t| \geq \eta\right] \leq 2e^{-2\eta^2 M_t},$$

$$\mathbb{P}\left[|\bar{X} - \mu| \geq \eta\right] \leq 2e^{-\frac{2\eta^2 M_t}{|\mathcal{V}|^2}},$$

$$\mathbb{P}\left[|\bar{X}_{t\backslash s} - \mu_t| \geq \eta\right] \leq 2e^{-2\eta^2 (M_t - 1)},$$

$$\mathbb{P}\left[|\bar{X}_{\backslash s} - \mu| \geq \eta\right] \leq 2e^{-\frac{2\eta^2 (M_t - 1)}{|\mathcal{V}|^2}}.$$

From this, we have:

$$0 < \mu_t - \eta \leq \bar{X}_t \leq \mu_t + \eta \quad \text{w.p. at least} \quad 1 - 2e^{-2\eta^2 M_t},$$

$$0 < \mu - \eta \leq \bar{X} \leq \mu + \eta \quad \text{w.p. at least} \quad 1 - 2e^{\frac{-2\eta^2 M_t}{|\mathcal{V}|^2}}$$

$$0 < \mu_t - \eta \leq \bar{X}_{t\backslash s} \leq \mu_t + \eta \quad \text{w.p. at least} \quad 1 - 2e^{-2\eta^2 (M_t - 1)},$$

$$0 < \mu - \eta \leq \bar{X}_{\backslash s} \leq \mu + \eta \quad \text{w.p. at least} \quad 1 - 2e^{\frac{-2\eta^2 (M_t - 1)}{|\mathcal{V}|^2}}.$$

Then, we have:

$$0 < \frac{\mu_t - \eta}{\mu + \eta} \leq \frac{\bar{X}_t}{\bar{X}} \leq \frac{\mu_t + \eta}{\mu - \eta} \quad \text{w.p. at least} \quad 1 - 2e^{-2\eta^2 M_t} - 2e^{\frac{-2\eta^2 M_t}{|\mathcal{V}|^2}},$$

$$0 < \frac{\mu_t - \eta}{\mu + \eta} \leq \frac{\bar{X}_{t\backslash s}}{\bar{X}_{\backslash s}} \leq \frac{\mu_t + \eta}{\mu - \eta} \quad \text{w.p. at least} \quad 1 - 2e^{-2\eta^2 (M_t - 1)} - 2e^{\frac{-2\eta^2 (M_t - 1)}{|\mathcal{V}|^2}}.$$

From the second result, we see that:

$$M_t \frac{\mu_t - \eta}{\mu + \eta} \leq M_t \frac{\bar{X}_t}{\bar{X}} \leq M_t \frac{\mu_t + \eta}{\mu - \eta} \quad \text{w.p at least} \quad 1 - 2\left[e^{-2\eta^2 M_t} + e^{\frac{-2\eta^2 M_t}{|\mathcal{V}|^2}}\right],$$

$$\sum_{s=1}^{M_t} \frac{\bar{X}_{t\backslash s}}{\bar{X}_{\backslash s}} \in \left[M_t \frac{\mu_t - \eta}{\mu + \eta}, M_t \frac{\mu_t + \eta}{\mu - \eta}\right] \quad \text{w.p at least} \quad 1 - 2M_t \left[e^{-2\eta^2 (M_t - 1)} + e^{\frac{-2\eta^2 (M_t - 1)}{|\mathcal{V}|^2}}\right].$$

Combining these two results, gives us:

$$M_t \frac{\bar{X}_t}{\bar{X}} - \frac{M_t - 1}{M_t} \sum_{s=1}^{M_t} \frac{\bar{X}_{t\backslash s}}{\bar{X}_{\backslash s}} \in \left[M_t \frac{\mu_t - \eta}{\mu + \eta} - (M_t - 1)\left[\frac{\mu_t + \eta}{\mu - \eta}\right], M_t \frac{\mu_t + \eta}{\mu - \eta} - (M_t - 1)\left[\frac{\mu_t - \eta}{\mu + \eta}\right]\right], \quad (9)$$

$$\text{w.p. at least} \quad 1 - 2M_t \left[e^{-2\eta^2 (M_t - 1)} + e^{\frac{-2\eta^2 (M_t - 1)}{|\mathcal{V}|^2}}\right] - 2\left[e^{-2\eta^2 M_t} + e^{\frac{-2\eta^2 M_t}{|\mathcal{V}|^2}}\right].$$

Let us study the boundaries in the above result. To do this, notice:

$$\frac{\mu_t + \eta}{\mu - \eta} = \frac{\mu_t}{\mu} + \eta \frac{\mu + \mu_t}{\mu(\mu - \eta)}, \quad \frac{\mu_t - \eta}{\mu + \eta} = \frac{\mu_t}{\mu} - \eta \frac{\mu + \mu_t}{\mu(\mu + \eta)}.$$

We start with the right hand side in Equation 9:

$$M_t \frac{\mu_t + \eta}{\mu - \eta} - (M_t - 1)\left[\frac{\mu_t - \eta}{\mu + \eta}\right] = M_t \left[\frac{\mu_t}{\mu} + \eta \frac{\mu + \mu_t}{\mu(\mu - \eta)}\right] - (M_t - 1)\left[\frac{\mu_t}{\mu} - \eta \frac{\mu + \mu_t}{\mu(\mu + \eta)}\right] \quad (10)$$

$$= \frac{\mu_t}{\mu} + M_t \eta \frac{\mu + \mu_t}{\mu(\mu - \eta)} + (M_t - 1)\eta \frac{\mu + \mu_t}{\mu(\mu + \eta)}$$

$$\leq \frac{\mu_t}{\mu} + \eta \frac{\mu + \mu_t}{\mu(\mu - \eta)} 2M_t + \eta \frac{\mu + \mu_t}{\mu(\mu + \eta)} 2M_t$$

$$= \frac{\mu_t}{\mu} + \eta \frac{4M_t(\mu + \mu_t)}{\mu^2 - \eta^2}.$$

Similarly, for the left hand side in Equation 9:

$$M_t \frac{\mu_t - \eta}{\mu + \eta} - (M_t - 1) \left[ \frac{\mu_t + \eta}{\mu - \eta} \right] = M_t \left[ \frac{\mu_t}{\mu} - \eta \frac{\mu + \mu_t}{\mu(\mu + \eta)} \right] - (M_t - 1) \left[ \frac{\mu_t}{\mu} + \eta \frac{\mu + \mu_t}{\mu(\mu - \eta)} \right]$$

$$= \frac{\mu_t}{\mu} - M_t \eta \frac{\mu + \mu_t}{\mu(\mu + \eta)} - (M_t - 1)\eta \frac{\mu + \mu_t}{\mu(\mu - \eta)}$$

$$\geq \frac{\mu_t}{\mu} - \eta \frac{\mu + \mu_t}{\mu(\mu + \eta)} 2M_t - \eta \frac{\mu + \mu_t}{\mu(\mu - \eta)} 2M_t$$

$$= \frac{\mu_t}{\mu} - \eta \frac{4M_t(\mu + \mu_t)}{\mu^2 - \eta^2}.$$

Notice, that $\eta \frac{4M_t(\mu+\mu_t)}{\mu^2-\eta^2} \leq \eta |\mathcal{V}|^T \mathcal{O}(1)$ Hence, denoting $c = \eta |\mathcal{V}|^T \mathcal{O}(1)$, using $\hat{p}_{\alpha, \text{JK}}^{(\text{pow})}(x_t|\boldsymbol{q}, \boldsymbol{x}_{0:t-1}) = M_t \frac{\bar{X}_t}{\bar{X}} - \frac{M_t - 1}{M_t} \sum_{s=1}^{M_t} \frac{\bar{X}_{t \setminus s}}{\bar{X}_{\setminus s}}$ and $p_\alpha^{(\text{pow})}(x_t|\boldsymbol{q}, \boldsymbol{x}_{0:t-1}) = \frac{\mu_t}{\mu}$ we have:

$$\hat{p}_{\alpha, \text{JK}}^{(\text{pow})}(x_t|\boldsymbol{q}, \boldsymbol{x}_{0:t-1}) \in \left[ p_\alpha^{(\text{pow})}(x_t|\boldsymbol{q}, \boldsymbol{x}_{0:t-1}) - c, p_\alpha^{(\text{pow})}(x_t|\boldsymbol{q}, \boldsymbol{x}_{0:t-1}) + c \right],$$

w.p. at least

$$1 - 2M_t \left[ e^{-2\eta^2(M_t-1)} + e^{\frac{-2\eta^2(M_t-1)}{|\mathcal{V}|^2}} \right] - 2 \left[ e^{-2\eta^2 M_t} + e^{\frac{-2\eta^2 M_t}{|\mathcal{V}|^2}} \right].$$

Or, equivalently:

$$|\hat{p}_{\alpha, \text{JK}}^{(\text{pow})}(x_t|\boldsymbol{q}, \boldsymbol{x}_{0:t-1}) - p_\alpha^{(\text{pow})}(x_t|\boldsymbol{q}, \boldsymbol{x}_{0:t-1})| \leq c,$$

w.p. at least

$$1 - 2M_t \left[ e^{-2\eta^2(M_t-1)} + e^{\frac{-2\eta^2(M_t-1)}{|\mathcal{V}|^2}} \right] - 2 \left[ e^{-2\eta^2 M_t} + e^{\frac{-2\eta^2 M_t}{|\mathcal{V}|^2}} \right].$$

Now, let us simplify the confidence probabilities:

$$1 - 2M_t \left[ e^{-2\eta^2(M_t-1)} + e^{\frac{-2\eta^2(M_t-1)}{|\mathcal{V}|^2}} \right] - 2 \left[ e^{-2\eta^2 M_t} + e^{\frac{-2\eta^2 M_t}{|\mathcal{V}|^2}} \right] \geq$$

$$1 - 2M_t \left[ e^{-2\eta^2(M_t-1)} + e^{\frac{-2\eta^2(M_t-1)}{|\mathcal{V}|^2}} \right] - 2M_t \left[ e^{-2\eta^2 M_t} + e^{\frac{-2\eta^2 M_t}{|\mathcal{V}|^2}} \right] \geq$$

$$1 - 4M_t \left[ e^{-2\eta^2(M_t-1)} + e^{\frac{-2\eta^2(M_t-1)}{|\mathcal{V}|^2}} \right] \geq 1 - 8M_t e^{\frac{-2\eta^2(M_t-1)}{|\mathcal{V}|^2}}.$$

So, in other words, we have:

$$\mathbb{P} \left[ |\hat{p}_{\alpha, \text{JK}}^{(\text{pow})}(x_t|\boldsymbol{q}, \boldsymbol{x}_{0:t-1}) - p_\alpha^{(\text{pow})}(x_t|\boldsymbol{q}, \boldsymbol{x}_{0:t-1})| \leq c \right] \geq 1 - 8M_t e^{\frac{-2\eta^2(M_t-1)}{|\mathcal{V}|^2}}.$$

Let $\epsilon = \eta |\mathcal{V}|^T \mathcal{O}(1)$, then $\eta = \frac{\epsilon}{|\mathcal{V}|^T} \mathcal{O}(1)$, then:

$$\mathbb{P} \left[ |\hat{p}_{\alpha, \text{JK}}^{(\text{pow})}(x_t|\boldsymbol{q}, \boldsymbol{x}_{0:t-1}) - p_\alpha^{(\text{pow})}(x_t|\boldsymbol{q}, \boldsymbol{x}_{0:t-1})| \leq \epsilon \right] \geq 1 - 8M_t e^{\frac{-\mathcal{O}(1)\epsilon^2(M_t-1)}{|\mathcal{V}|^{2T+2}}}.$$

We need the confidence bound to be at least $1 - \delta$ for a given $\delta$, hence:

$$1 - 8M_t e^{\frac{-\mathcal{O}(1)\epsilon^2(M_t-1)}{|\mathcal{V}|^{2T+2}}} \geq 1 - \delta \implies 8M_t e^{\frac{-\mathcal{O}(1)\epsilon^2(M_t-1)}{|\mathcal{V}|^{2T+2}}} \leq \delta \implies e^{\frac{-\mathcal{O}(1)\epsilon^2(M_t-1)}{|\mathcal{V}|^{2T+2}}} \leq \frac{\delta}{8|\mathcal{V}|^T}$$

$$\implies \frac{\mathcal{O}(1)\epsilon^2(M_t-1)}{|\mathcal{V}|^{2T+2}} \geq \log \left( \frac{|\mathcal{V}|^T}{\delta} \right) \implies M_t \geq \mathcal{O} \left( \frac{1}{\epsilon^2} \log \left( \frac{1}{\delta} \right) \right).$$

This finishes the proof of the claim. $\qquad\square$

## A.5. Explicit Constants in the Bias Bounds

We derive explicit constants hidden in the big-O notation $\mathcal{O}(1/M_t)$ and $\mathcal{O}(1/M_t^2)$ in Lemmas 3.2 and 3.3 under a worst-case scenario, where all possible rollouts $\boldsymbol{x}_{t:T}$ of length $T - t + 1$ are assumed to be equally likely under the LLM. This assumption is generally unrealistic for pretrained LLMs, and the resulting constants can be quite loose in practice.

Let us start with Lemma 3.2. Notice that, for the second-order Taylor expansion of $(1 + z)^{-1}$, the following result appears in Eq. (8) in the proof of Lemma 3.2:

$$\hat{p}_\alpha^{(\text{pow})}(x_t|\boldsymbol{q}, \boldsymbol{x}_{0:t-1}) \approx \frac{\zeta_t(x_t)p^\alpha(x_t|\boldsymbol{q}, \boldsymbol{x}_{0:t-1})}{\sum_{x_t' \in \mathcal{V}} \zeta_t(x_t')p^\alpha(x_t'|\boldsymbol{q}, \boldsymbol{x}_{0:t-1})} + \underbrace{\frac{\delta(x_t)}{\sum_{x_t' \in \mathcal{V}} \zeta_t(x_t')p^\alpha(x_t'|\boldsymbol{q}, \boldsymbol{x}_{0:t-1})}}_{\mathcal{A}} \tag{11}$$

$$- \underbrace{\frac{\zeta_t(x_t)p^\alpha(x_t|\boldsymbol{q}, \boldsymbol{x}_{0:t-1}) \sum_{x_t' \in \mathcal{V}} \delta(x_t')}{\left[\sum_{x_t' \in \mathcal{V}} \zeta_t(x_t')p^\alpha(x_t'|\boldsymbol{q}, \boldsymbol{x}_{0:t-1})\right]^2}}_{\mathcal{B}} - \underbrace{\frac{\delta(x_t) \sum_{x_t' \in \mathcal{V}} \delta(x_t')}{\left[\sum_{x_t' \in \mathcal{V}} \zeta_t(x_t')p^\alpha(x_t'|\boldsymbol{q}, \boldsymbol{x}_{0:t-1})\right]^2}}_{\mathcal{C}}$$

$$+ \underbrace{\frac{\zeta_t(x_t)p^\alpha(x_t|\boldsymbol{q}, \boldsymbol{x}_{0:t-1}) \left[\sum_{x_t' \in \mathcal{V}} \delta(x_t')\right]^2}{\left[\sum_{x_t' \in \mathcal{V}} \zeta_t(x_t')p^\alpha(x_t'|\boldsymbol{q}, \boldsymbol{x}_{0:t-1})\right]^3}}_{\mathcal{D}}.$$

Here

$$\delta(x_t) = \frac{1}{M_t} \sum_{r=1}^{M_t} p^\alpha(x_t|\boldsymbol{q}, \boldsymbol{x}_{0:t-1}) \left[p^{\alpha-1}(\boldsymbol{x}_{t+1:T}^{[r]}|\boldsymbol{q}, \boldsymbol{x}_{0:t}) - \mathbb{E}_{\boldsymbol{x}_{t+1:T} \sim p(\cdot|\boldsymbol{q}, \boldsymbol{x}_{0:t})} \left[p^{\alpha-1}(\boldsymbol{x}_{t+1:T}|\boldsymbol{q}, \boldsymbol{x}_{0:t})\right]\right].$$

From this we have $\mathbb{E}[\delta(x_t)] = 0$ for any $x_t \in \mathcal{V}$, where the expectation is taken with respect to the randomness associated with terms $\delta(x_t')$, $x_t' \in \mathcal{V}$. This immediately gives $\mathbb{E}[\mathcal{A}] = \mathbb{E}[\mathcal{B}] = 0$. Now, let us take a closer look at the other terms. To do so, we need to lower-bound the term in the denominators of these expressions. We rewrite this expression as

$$\sum_{x_t \in \mathcal{V}} \zeta_t(x_t)p^\alpha(x_t|\boldsymbol{q}, \boldsymbol{x}_{0:t-1}) = \sum_{x_t \in \mathcal{V}} \sum_{\boldsymbol{x}_{t+1:T}} p^\alpha(\boldsymbol{x}_{t+1:T}|\boldsymbol{q}, \boldsymbol{x}_{0:t-1}, x_t)p^\alpha(x_t|\boldsymbol{q}, \boldsymbol{x}_{0:t-1})$$

$$= \sum_{\boldsymbol{x}_{t:T}} p^\alpha(\boldsymbol{x}_{t:T}|\boldsymbol{q}, \boldsymbol{x}_{0:t-1}) = \sum_{i=1}^{A_t} p_i^\alpha,$$

where we list all possible continuations $\{\boldsymbol{x}_{t:T}\}$ of length $T - t + 1$ from vocabulary $\mathcal{V}$, and $p_i$ is the probability of the $i$-th continuation from this list, i.e., $p_i = p(i\text{-th continuation}|\boldsymbol{q}, \boldsymbol{x}_{0:t-1})$. It is easy to see that the total number of such continuations is $A_t = |\mathcal{V}|^{T-t+1}$. Hence, to lower-bound the expression $\sum_{i=1}^{A_t} p_i^\alpha$, we cast the following optimization problem:

$$\min \sum_{i=1}^{A_t} p_i^\alpha$$

$$\text{s.t.} \quad \sum_{i=1}^{A_t} p_i = 1, \qquad p_i \geq 0 \quad \forall i = 1, \ldots, A_t.$$

It is easy to see, by constructing the Lagrangian, that the minimal value of the above expression for $\alpha > 1$ is achieved when $p_1 = \cdots = p_{A_t} = 1/A_t$, and is given as

$$\sum_{x_t \in \mathcal{V}} \zeta_t(x_t)p^\alpha(x_t|\boldsymbol{q}, \boldsymbol{x}_{0:t-1}) \geq \frac{1}{A_t^{\alpha-1}} = \frac{1}{|\mathcal{V}|^{(T-t+1)(\alpha-1)}} \implies \tag{12}$$

$$\frac{1}{\sum_{x_t \in \mathcal{V}} \zeta_t(x_t)p^\alpha(x_t|\boldsymbol{q}, \boldsymbol{x}_{0:t-1})} \leq |\mathcal{V}|^{(T-t+1)(\alpha-1)}.$$

Equipped with this bound, we can proceed to upper-bound other terms in Equation 11:

$$\mathbb{E}[\mathcal{C}] = \mathbb{E}\left[\frac{\delta(x_t)\sum_{x_t'\in\mathcal{V}}\delta(x_t')}{\left[\sum_{x_t'\in\mathcal{V}}\zeta_t(x_t')p^\alpha(x_t'|\boldsymbol{q},\boldsymbol{x}_{0:t-1})\right]^2}\right] = \frac{\mathbb{E}[\delta^2(x_t)]}{\left[\sum_{x_t'\in\mathcal{V}}\zeta_t(x_t')p^\alpha(x_t'|\boldsymbol{q},\boldsymbol{x}_{0:t-1})\right]^2} \tag{13}$$

$$\leq |\mathcal{V}|^{2(T-t+1)(\alpha-1)}\mathbb{E}[\delta^2(x_t)] \leq \frac{4|\mathcal{V}|^{2(T-t+1)(\alpha-1)}}{M_t}.$$

From the above analysis we have

$$\mathbb{E}[\delta^2(x_t)] \leq \frac{4}{M_t}.$$

Similarly,

$$\mathbb{E}[\mathcal{D}] = \mathbb{E}\left[\frac{\zeta_t(x_t)p^\alpha(x_t|\boldsymbol{q},\boldsymbol{x}_{0:t-1})\left[\sum_{x_t'\in\mathcal{V}}\delta(x_t')\right]^2}{\left[\sum_{x_t'\in\mathcal{V}}\zeta_t(x_t')p^\alpha(x_t'|\boldsymbol{q},\boldsymbol{x}_{0:t-1})\right]^3}\right]$$

$$\leq |\mathcal{V}|^{3(T-t+1)(\alpha-1)}\mathbb{E}\left[\zeta_t(x_t)p^\alpha(x_t|\boldsymbol{q},\boldsymbol{x}_{0:t-1})\left[\sum_{x_t'\in\mathcal{V}}\delta(x_t')\right]^2\right]$$

$$\leq |\mathcal{V}|^{3(T-t+1)(\alpha-1)}|\mathcal{V}|\mathbb{E}\left[\delta^2(x_t)\right] \leq \frac{4|\mathcal{V}|^{3(T-t+1)(\alpha-1)+1}}{M_t}.$$

Finally, combining these results, we have:

$$\mathbb{E}\left[\hat{p}_\alpha^{(\text{pow})}(x_t|\boldsymbol{q},\boldsymbol{x}_{0:t-1})\right] - \frac{\zeta_t(x_t)p^\alpha(x_t|\boldsymbol{q},\boldsymbol{x}_{0:t-1})}{\sum_{x_t'\in\mathcal{V}}\zeta_t(x_t')p^\alpha(x_t'|\boldsymbol{q},\boldsymbol{x}_{0:t-1})} \tag{14}$$

$$\leq \frac{4|\mathcal{V}|^{2(T-t+1)(\alpha-1)}}{M_t}\left[1 + |\mathcal{V}|^{2(T-t+1)(\alpha-1)+1}\right].$$

Note that this analysis corresponds to a worst-case scenario, where all possible rollouts $\boldsymbol{x}_{t:T}$ of length $T-t+1$ are assumed to be equally likely under the LLM. In practice, this assumption is unrealistic for modern pretrained models, which typically assign highly non-uniform probabilities to different rollouts. To derive more meaningful and tighter bounds, it is necessary to introduce assumptions about the rollout distribution induced by the LLM. Such assumptions would enable a more accurate characterization of the denominator term.

Now, let us move to Lemma 3.3. Since this result is based on Lemma 3.2, in order to get the constant factor for the $\mathcal{O}(1/M_t^2)$ term, the Taylor expansion for $(1+z)^{-1}$ in Eq. (8) in the proof of Lemma 3.2 should be written to the fourth order, since such a decomposition provides the exact form of the coefficient $b_2$ in the proof of Lemma 3.3:

$$\hat{p}_\alpha^{(\text{pow})}(x_t|\boldsymbol{q},\boldsymbol{x}_{0:t-1}) = \frac{\zeta_t(x_t)p^\alpha(x_t|\boldsymbol{q},\boldsymbol{x}_{0:t-1}) + \delta(x_t)}{\sum_{x_t'\in\mathcal{V}}\zeta_t(x_t')p^\alpha(x_t'|\boldsymbol{q},\boldsymbol{x}_{0:t-1})}\left[1 + \underbrace{\frac{\sum_{x_t'\in\mathcal{V}}\delta(x_t')}{\sum_{x_t'\in\mathcal{V}}\zeta_t(x_t')p^\alpha(x_t'|\boldsymbol{q},\boldsymbol{x}_{0:t-1})}}_{z}\right]^{-1}$$

$$= \frac{\zeta_t(x_t)p^\alpha(x_t|\boldsymbol{q},\boldsymbol{x}_{0:t-1}) + \delta(x_t)}{\sum_{x_t'\in\mathcal{V}}\zeta_t(x_t')p^\alpha(x_t'|\boldsymbol{q},\boldsymbol{x}_{0:t-1})}\left[1 - z + z^2 - z^3 + z^4 + \mathcal{O}(z^5)\right]$$

$$= \frac{\zeta_t(x_t)p^\alpha(x_t|\boldsymbol{q},\boldsymbol{x}_{0:t-1})}{\sum_{x_t'\in\mathcal{V}}\zeta_t(x_t')p^\alpha(x_t'|\boldsymbol{q},\boldsymbol{x}_{0:t-1})} + \underbrace{\frac{\zeta_t(x_t)p^\alpha(x_t|\boldsymbol{q},\boldsymbol{x}_{0:t-1})}{\sum_{x_t'\in\mathcal{V}}\zeta_t(x_t')p^\alpha(x_t'|\boldsymbol{q},\boldsymbol{x}_{0:t-1})}\left[-z + z^2 - z^3 + z^4\right]}_{\mathcal{A}_1}$$

$$+ \underbrace{\frac{\delta(x_t)}{\sum_{x_t'\in\mathcal{V}}\zeta_t(x_t')p^\alpha(x_t'|\boldsymbol{q},\boldsymbol{x}_{0:t-1})}\left[1 - z + z^2 - z^3 + z^4\right]}_{\mathcal{A}_2} + \mathcal{O}(z^5).$$

Terms $\mathcal{O}(z^5)$ do not contribute to the coefficient $b_2$ of interest. Hence, we focus on the above two terms. Let us start with $\mathcal{A}_1$. To simplify the notation, let us introduce

$$Y_r(x_t) = p^\alpha(x_t|\boldsymbol{q}, \boldsymbol{x}_{0:t-1}) \left[ p^{\alpha-1}(\boldsymbol{x}_{t+1:T}^{[r]}|\boldsymbol{q}, \boldsymbol{x}_{0:t}) - \mathbb{E}_{\boldsymbol{x}_{t+1:T} \sim p(\cdot|\boldsymbol{q}, \boldsymbol{x}_{0:t})} \left[ p^{\alpha-1}(\boldsymbol{x}_{t+1:T}|\boldsymbol{q}, \boldsymbol{x}_{0:t}) \right] \right],$$

$$\text{so} \quad \mathbb{E}[Y_r(x_t)] = 0, \qquad \mathbb{E}[|Y_r(x_t)|^k] \leq 2^k, \qquad \text{for } k \in \mathbb{N}.$$

Then, we have $\delta(x_t) = \frac{1}{M_t} \sum_{r=1}^{M_t} Y_r(x_t)$ and immediately:

$$\mathbb{E}\left[ \sum_{x_t \in \mathcal{V}} \delta(x_t) \right] = 0, \tag{15}$$

$$\mathbb{E}\left[ \left( \sum_{x_t \in \mathcal{V}} \delta(x_t) \right)^2 \right] \leq \frac{4|\mathcal{V}|}{M_t},$$

$$\mathbb{E}\left[ \left( \sum_{x_t \in \mathcal{V}} \delta(x_t) \right)^3 \right] = \frac{|\mathcal{V}|}{M_t^3} M_t \mathbb{E}[|Y_r(x_t)|^3] \leq \frac{8|\mathcal{V}|}{M_t^2},$$

$$\mathbb{E}\left[ \left( \sum_{x_t \in \mathcal{V}} \delta(x_t) \right)^4 \right] = \mathbb{E}\left[ \sum_{x_t \in \mathcal{V}} \delta^4(x_t) \right] + 6\mathbb{E} \sum_{t<t'} \left[ \delta^2(x_t)\delta^2(x_t') \right]$$

$$\leq |\mathcal{V}|\mathbb{E}\left[ \delta^4(x_t) \right] + \frac{48}{M_t^2}|\mathcal{V}|(|\mathcal{V}| - 1)$$

$$\leq |\mathcal{V}| \left[ \frac{4}{M_t^3} + \frac{12}{M_t^2} \right] + \frac{48}{M_t^2}|\mathcal{V}|(|\mathcal{V}| - 1) = \frac{4|\mathcal{V}|}{M_t^3} + \frac{12|\mathcal{V}|}{M_t^2} \left[ 4|\mathcal{V}| - 3 \right],$$

where we used that random variables $\delta(x_t)$ and $\delta(x_t')$ are independent when $x_t \neq x_t'$. Given these results, using Equation 12, we have

$$\mathbb{E}[z] = 0,$$

$$\mathbb{E}[z^2] = \frac{\mathbb{E}\left[ \left( \sum_{x_t' \in \mathcal{V}} \delta(x_t') \right)^2 \right]}{\left[ \sum_{x_t' \in \mathcal{V}} \zeta_t(x_t') p^\alpha(x_t'|\boldsymbol{q}, \boldsymbol{x}_{0:t-1}) \right]^2} \leq \frac{4|\mathcal{V}|}{M_t}|\mathcal{V}|^{2(T-t+1)(\alpha-1)},$$

$$\mathbb{E}[z^3] = \frac{\mathbb{E}\left[ \left( \sum_{x_t' \in \mathcal{V}} \delta(x_t') \right)^3 \right]}{\left[ \sum_{x_t' \in \mathcal{V}} \zeta_t(x_t') p^\alpha(x_t'|\boldsymbol{q}, \boldsymbol{x}_{0:t-1}) \right]^3} \leq \frac{8|\mathcal{V}|}{M_t^2}|\mathcal{V}|^{3(T-t+1)(\alpha-1)},$$

$$\mathbb{E}[z^4] = \frac{\mathbb{E}\left[ \left( \sum_{x_t' \in \mathcal{V}} \delta(x_t') \right)^4 \right]}{\left[ \sum_{x_t' \in \mathcal{V}} \zeta_t(x_t') p^\alpha(x_t'|\boldsymbol{q}, \boldsymbol{x}_{0:t-1}) \right]^4}$$

$$\leq \left[ \frac{4|\mathcal{V}|}{M_t^3} + \frac{12|\mathcal{V}|}{M_t^2} \left[ 4|\mathcal{V}| - 3 \right] \right] |\mathcal{V}|^{4(T-t+1)(\alpha-1)}.$$

Hence, for $\mathcal{A}_1$, we have:

$$\mathbb{E}\left[ \mathcal{A}_1 \right] \leq |\mathcal{V}|^{(T-t+1)(\alpha-1)} \mathbb{E}\left[ -z + z^2 - z^3 + z^4 \right]$$

$$\leq |\mathcal{V}|^{(T-t+1)(\alpha-1)} \left[ \frac{4|\mathcal{V}|}{M_t}|\mathcal{V}|^{2(T-t+1)(\alpha-1)} - \frac{8|\mathcal{V}|}{M_t^2}|\mathcal{V}|^{3(T-t+1)(\alpha-1)} \right.$$

$$\left. + \left[ \frac{4|\mathcal{V}|}{M_t^3} + \frac{12|\mathcal{V}|}{M_t^2} \left[ 4|\mathcal{V}| - 3 \right] \right] |\mathcal{V}|^{4(T-t+1)(\alpha-1)} \right]$$

$$= \frac{4|\mathcal{V}|^{3(T-t+1)(\alpha-1)+1}}{M_t^2} \left[ 3|\mathcal{V}|^{(T-t+1)(\alpha-1)}[4|\mathcal{V}| - 3] - 2 \right] + \text{rest terms.}$$

Because we are focused on the coefficient $b_2$ in the proof of Lemma 3.3, we are interested only in the coefficient associated with $1/M_t^2$. Thus, we have computed such a term for $\mathcal{A}_1$. Next, we focus on $\mathcal{A}_2$. Notice:

$$\mathbb{E}[\delta(x_t)z] \leq \frac{4|\mathcal{V}|^{(T-t+1)(\alpha-1)}}{M_t}, \tag{16}$$

$$\mathbb{E}[\delta(x_t)z^2] = \frac{\mathbb{E}\left[\delta(x_t)\left[\sum_{x_t'\in\mathcal{V}}\delta(x_t')\right]^2\right]}{\left[\sum_{x_t'\in\mathcal{V}}\zeta_t(x_t')p^\alpha(x_t'|\boldsymbol{q},\boldsymbol{x}_{0:t-1})\right]^2} = \frac{\mathbb{E}[\delta^3(x_t)]}{\left[\sum_{x_t'\in\mathcal{V}}\zeta_t(x_t')p^\alpha(x_t'|\boldsymbol{q},\boldsymbol{x}_{0:t-1})\right]^2} \leq \frac{8|\mathcal{V}|^{2(T-t+1)(\alpha-1)}}{M_t^2},$$

$$\mathbb{E}[\delta(x_t)z^3] = \frac{\mathbb{E}\left[\delta(x_t)\left[\sum_{x_t'\in\mathcal{V}}\delta(x_t')\right]^3\right]}{\left[\sum_{x_t'\in\mathcal{V}}\zeta_t(x_t')p^\alpha(x_t'|\boldsymbol{q},\boldsymbol{x}_{0:t-1})\right]^3} = \frac{\mathbb{E}\left[\delta^4(x_t)\right]}{\left[\sum_{x_t'\in\mathcal{V}}\zeta_t(x_t')p^\alpha(x_t'|\boldsymbol{q},\boldsymbol{x}_{0:t-1})\right]^3}$$

$$\leq \left[\frac{4}{M_t^3} + \frac{12}{M_t^2}\right]|\mathcal{V}|^{3(T-t+1)(\alpha-1)},$$

$$\mathbb{E}[\delta(x_t)z^4] = \frac{\mathbb{E}\left[\delta(x_t)\left[\sum_{x_t'\in\mathcal{V}}\delta(x_t')\right]^4\right]}{\left[\sum_{x_t'\in\mathcal{V}}\zeta_t(x_t')p^\alpha(x_t'|\boldsymbol{q},\boldsymbol{x}_{0:t-1})\right]^4} = \text{does not have terms of order } \frac{1}{M_t^2}.$$

So, this gives for the coefficient associated with $1/M_t^2$ in $\mathcal{A}_2$ the following expression:

$$\mathbb{E}[\mathcal{A}_2] \leq \frac{4|\mathcal{V}|^{2(T-t+1)(\alpha-1)}}{M_t^2}\left[2 - 3|\mathcal{V}|^{(T-t+1)(\alpha-1)}\right] + \text{rest terms}.$$

Combining the results for $\mathcal{A}_1$ and $\mathcal{A}_2$ gives, letting $w = |\mathcal{V}|^{(T-t+1)(\alpha-1)}$,

$$\mathbb{E}\left[\hat{p}_\alpha^{(\text{pow})}(x_t|\boldsymbol{q},\boldsymbol{x}_{0:t-1})\right] \leq \left[\frac{4w^3|\mathcal{V}|}{M_t^2}[3w[4|\mathcal{V}|-3]-2] + \frac{4w^2}{M_t^2}[2-3w]\right] + \text{rest terms}$$

$$= \underbrace{4w^2\left[12w^2|\mathcal{V}|^2 - 9w^2|\mathcal{V}| - 2w|\mathcal{V}| + 2 - 3w\right]}_{b_2}\frac{1}{M_t^2} + \text{rest terms}.$$

Following the proof of Lemma 3.3, the expression $b_2$ gives the constant factor in front of the $1/M_t^2$ term.

Again, this analysis corresponds to a worst-case scenario, where all possible rollouts $\boldsymbol{x}_{t:T}$ of length $T - t + 1$ are assumed to be equally likely under the LLM. In practice, this assumption is unrealistic for modern pretrained models, which typically assign highly non-uniform probabilities to different rollouts. To derive more meaningful and tighter bounds, it is necessary to introduce assumptions about the rollout distribution induced by the LLM. Such assumptions would enable a more accurate characterization of the denominator term.

## B. Batch Tokens Extension

In Algorithm 1, we provide a batched version of the algorithm presented in the main paper. Next, we provide a detailed description of the main steps in this algorithm and further discuss its advantages. To proceed in batches of size $B$ over trajectories with length $L$, we first identify the total number of iterations in the main for loop as $\lfloor\frac{T}{B}\rfloor$, and the remaining part of the trajectory is of size $q = T \mod B$. Next, in each iteration $t$, based on the trajectory generated so far $\boldsymbol{x}_{0:(t-1)B-1}$[1], we aim to sample the next chunk $\boldsymbol{x}_{(t-1)B:tB-1}$ of $B$ tokens from our approximation of the power distribution. To achieve this, we first generate a collection of $L_t$ candidates for the next chunk $\boldsymbol{x}_{(t-1)B:tB-1}$. We call this collection $\mathcal{L}_t$ (see line 4) and its elements are sampled from the base model, i.e., $\boldsymbol{x}_{(t-1)B:tB-1}^{[r]} \sim p(\cdot|\boldsymbol{q},\boldsymbol{x}_{0:(t-1)B-1})$ for $r = 1,\ldots L_t$. Next, from the collection $\mathcal{L}_t$ we pick $K_t$, the most promising candidates, based on their likelihood provided by the base model. In line 5, these most promising candidates are aggregated in the set $\mathcal{G}_t(\boldsymbol{x}_{0:(t-1)B-1})$. In lines 6-9 we start the first inner **for loop**

---

[1]When $t = 1$ we have $(t-1)B - 1 = -1$ and we use the notation $\boldsymbol{x}_{0:-1} = \{\}$

---

**Algorithm 2** Batch Version of Scalable Power Sampling for a Query $q$

---

1: **Inputs:** Power distribution exponent $\alpha \in [1, \infty)$, upper bounds on trajectory length $T$, size of a batch $B$, size of groups $\{K_t\}_{t=1}^{\lfloor \frac{T}{B} \rfloor + 1}$, size of top evaluation budgets $\{L_t\}_{t=1}^{\lfloor \frac{T}{B} \rfloor + 1}$, horizon lengths $\{H_t\}_{t=1}^{\lfloor \frac{T}{B} \rfloor}$, and the total Monte Carlo budgets $\{M_t\}_{t=1}^{\lfloor \frac{T}{B} \rfloor}$

2: Let $q = T \mod B$ and $w = \lfloor \frac{T}{B} \rfloor + 1$

3: **for** $t = 1 \rightarrow \lfloor \frac{T}{B} \rfloor$ **do**

4:    Sample $L_t$ blocks of size $B$: $\mathcal{L}_t = \{\{x_{(t-1)B:tB-1}^{[r]}\}_{r=1}^{L_t}$ such that $x_{(t-1)B:tB-1}^{[r]} \sim p(\cdot|q, x_{0:(t-1)B-1})\}$

5:    Compute $\mathcal{G}_t(x_{0:(t-1)B-1}) = \text{Top@}K_t \left[ \text{in } \mathcal{L}_t \text{ with respect to } p(\cdot|q, x_{0:(t-1)B-1}) \right]$

6:    **for** $x'_{(t-1)B:tB-1} \in \mathcal{G}_t(x_{0:(t-1)B-1})$ **do**

7:      Sample $M_t$ i.i.d. completions of length $H_t$:

$$\{\{x_{tB:tB+H_t-1}^{[r]}\}_{r=1}^{M_t} \text{ such that } x_{tB:tB+H_t-1}^{[r]} \sim p(\cdot|q, x_{0:(t-1)B-1}, x'_{(t-1)B:tB-1})\}$$

8:      Compute $\hat{\zeta}_t(x'_{(t-1)B:tB-1})$ and $\hat{\zeta}_{t,-s}^{(\text{LOO})}(x'_{(t-1)B:tB-1})$ from Eq. 17 and 18

9:    **end for**

10:    **for** $x'_{(t-1)B:tB-1} \in \mathcal{G}_t(x_{0:(t-1)B-1})$ **do**

11:      **for** $s \in \{1, \ldots, M_t\}$ **do**

12:        Compute $\hat{p}_\alpha^{(\text{pow})}(x'_{(t-1)B:tB-1})$ and $\hat{p}_{\alpha,-s}^{(\text{pow})}(x'_{(t-1)B:tB-1})$ using Equations 19 and 20

13:      **end for**

14:    **end for**

15:    Compute the Jackknife estimator using Equation 21

16:    Sample $x_{(t-1)B:tB-1} \in \mathcal{G}_t(x_{0:(t-1)B-1})$ w.p. $\hat{p}_{\alpha,\text{ JK}}^{(\text{pow})}(x_{(t-1)B:tB-1}|q, x_{0:(t-1)B-1})$

17:    Concatenate the sampled tokens $x_{0:tB-1} = x_{0:(t-1)B-1} \oplus x_{(t-1)B:tB-1}$

18: **end for**

19: Sample $L_w$ blocks of size $q+1$: $\mathcal{L}_w = \{\{x_{\lfloor \frac{T}{B} \rfloor B:T}^{[r]}\}_{r=1}^{L_w}$ such that $x_{\lfloor \frac{T}{B} \rfloor B:T}^{[r]} \sim p(\cdot|q, x_{0:B\lfloor \frac{T}{B} \rfloor - 1})\}$

20: Compute $\mathcal{G}_w(x_{0:B\lfloor \frac{T}{B} \rfloor - 1}) = \text{Top@}K_w \left[ \text{in } \mathcal{L}_w \text{ with respect to } p(\cdot|q, x_{0:B\lfloor \frac{T}{B} \rfloor - 1}) \right]$

21: Sample $x_{\lfloor \frac{T}{B} \rfloor B:T} \in \mathcal{G}_w(x_{0:B\lfloor \frac{T}{B} \rfloor - 1})$ w. p. $\hat{p}_\alpha^{(\text{low.temp})}(x_{\lfloor \frac{T}{B} \rfloor B:T}|q, x_{0:B\lfloor \frac{T}{B} \rfloor - 1})$ given in Equation 22

22: Concatenate the sampled token $x_{0:T} = x_{0:B\lfloor \frac{T}{B} \rfloor - 1} \oplus x_{\lfloor \frac{T}{B} \rfloor B:T}$

23: **Return** Sampled trajectory $\{x_0, \ldots, x_T\} = x_{0:T}$

---

that aims to compute the following scaling factors for each candidate chunk $x'_{(t-1)B:tB-1} \in \mathcal{G}_t(x_{0:(t-1)B-1})$:

$$\hat{\zeta}_t(x'_{(t-1)B:tB-1}) = \frac{1}{M_t} \sum_{r=1}^{M_t} p^{\alpha-1}(x_{tB:tB+H_t-1}^{[r]}|q, x_{0:(t-1)B-1}, x'_{(t-1)B:tB-1}) \tag{17}$$

$$\hat{\zeta}_{t,-s}^{(\text{LOO})}(x'_{(t-1)B:tB-1}) = \frac{1}{M_t - 1} \sum_{r=1,r\neq s}^{M_t} p^{\alpha-1}(x_{tB:tB+H_t-1}^{[r]}|q, x_{0:(t-1)B-1}, x'_{(t-1)B:tB-1}), \tag{18}$$

where each roll-out trajectory $x_{tB:tB+H_t-1}^{[r]}$ has length $H_t$ tokens and is sampled from the base model (line 7). In lines 10-14 we start the second inner **for loop** that compute standard and leave-one-out probabilities:

$$\hat{p}_\alpha^{(\text{pow})}(x'_{(t-1)B:tB-1}) = \frac{p^\alpha(x'_{(t-1)B:tB-1}|q, x_{0:(t-1)B-1})\hat{\zeta}_t(x'_{(t-1)B:tB-1})}{\sum_{x_{(t-1)B:tB-1} \in \mathcal{G}_t(x_{0:(t-1)B-1})} p^\alpha(x_{(t-1)B:tB-1}|q, x_{0:(t-1)B-1})\hat{\zeta}_t(x_{(t-1)B:tB-1})}. \tag{19}$$

$$\hat{p}_{\alpha,-s}^{(\text{pow})}(x'_{(t-1)B:tB-1}) = \frac{p^\alpha(x'_{(t-1)B:tB-1}|q, x_{0:(t-1)B-1})\hat{\zeta}_{t,-s}^{(\text{LOO})}(x'_{(t-1)B:tB-1})}{\sum_{x_{(t-1)B:tB-1} \in \mathcal{G}_t(x_{0:(t-1)B-1})} p^\alpha(x_{(t-1)B:tB-1}|q, x_{0:(t-1)B-1})\hat{\zeta}_{t,-s}^{(\text{LOO})}(x_{(t-1)B:tB-1})}. \tag{20}$$

for all $s \in \{1, \ldots, M_t\}$. Being equipped with these probabilities, for each candidate block $\boldsymbol{x}'_{(t-1)B:tB-1} \in \mathcal{G}_t(\boldsymbol{x}_{0:(t-1)B-1})$, we compute jackknife likelihoods in line 15 using the following equation:

$$\hat{p}^{(\text{pow})}_{\alpha, \text{JK}}(\boldsymbol{x}'_{(t-1)B:tB-1}|\boldsymbol{q}, \boldsymbol{x}_{0:(t-1)B-1}) = M_t \hat{p}^{(\text{pow})}_{\alpha}(\boldsymbol{x}'_{(t-1)B:tB-1}) - \frac{M_t - 1}{M_t} \sum_{s=1}^{M_t} \hat{p}^{(\text{pow})}_{\alpha,-s}(\boldsymbol{x}'_{(t-1)B:tB-1}). \tag{21}$$

In lines 16-17, we sample the next block in the trajectory using the computed jackknife probability distribution and concatenate this block to the already constructed prefix. Finally, in lines 19-20 we construct the last part of the trajectory of length $q + 1 = T - B\lfloor \frac{T}{B} \rfloor + 1$ by generating a collection of $L_w$ candidate completions $\boldsymbol{x}^{[r]}_{\lfloor \frac{T}{B} \rfloor B:T} \sim p(\cdot|\boldsymbol{q}, \boldsymbol{x}_{0:B\lfloor \frac{T}{B} \rfloor - 1})\}$ (see line 19). In line 20, we select $K_w$, the most promising completions from group $\mathcal{L}_w$, with respect to their likelihood values. Then, we sample one of these promising candidates, using a batch version of low temperature distribution (line 21):

$$\hat{p}^{(\text{low.temp})}_{\alpha}(\boldsymbol{x}_{\lfloor \frac{T}{B} \rfloor B:T}|\boldsymbol{q}, \boldsymbol{x}_{0:B\lfloor \frac{T}{B} \rfloor - 1}) = \frac{p^{\alpha}(\boldsymbol{x}_{\lfloor \frac{T}{B} \rfloor B:T}|\boldsymbol{q}, \boldsymbol{x}_{0:B\lfloor \frac{T}{B} \rfloor - 1})}{\sum_{\hat{\boldsymbol{x}}_{\lfloor \frac{T}{B} \rfloor B:T} \in \mathcal{G}_w(\boldsymbol{x}_{0:B\lfloor \frac{T}{B} \rfloor - 1})} p^{\alpha}(\hat{\boldsymbol{x}}_{\lfloor \frac{T}{B} \rfloor B:T}|\boldsymbol{q}, \boldsymbol{x}_{0:B\lfloor \frac{T}{B} \rfloor - 1})}. \tag{22}$$

The selected completion is concatenated to the constructed trajectory (line 22) and forms the output of the algorithm.

Please notice, in Algorithm 2 each generation of future rollouts (lines 4,7 and 19) can be performed in a parallel fashion, thus reducing further the overall processing time. The overall inference complexity of the above algorithm is given by $\sum_{t=1}^{\lfloor \frac{T}{B} \rfloor}(L_t + K_t M_t) + L_w$. We can also derive such bounds for the original algorithm in the main paper in terms of token generation complexity. This amounts to: $\mathcal{O}\left(\sum_{t=1}^{\lfloor \frac{T}{B} \rfloor}[L_t B + K_t M_t H_t] + L_w B\right)$.

### B.1. Induced Bias of Batch Tokens Extension

The introduction of the truncation parameter $K_t$ and the lookahead horizon $H_t$ induces additional bias relative to the true power distribution. In general, this bias is difficult to quantify without imposing further assumptions on the LLM conditional distribution $p(\cdot|\boldsymbol{q}, \boldsymbol{x}_{0:t})$. One possible approach is to assume a specific asymptotic behavior of $p(\cdot|\boldsymbol{q}, \boldsymbol{x}_{0:t})$ as a function of the prefix length $t$, which would enable us to characterize the impact of the lookahead parameter $H_t$. Similarly, by introducing assumptions on how the probability mass decays from the most likely tokens to less likely ones, we can quantify the effect of truncating the distribution to the top $K_t$ tokens.

Let us consider some strategies we can suggest here. Let us start with the truncation parameter $K_t$. It is defined as the subset of tokens in the vocabulary $\mathcal{V}$ with model probability exceeding a threshold $\kappa$, then the resulting bias can be expressed as a function of $\kappa$. In particular, let us assume that we restrict our attention to the tokens $\mathcal{V}(\kappa) \subseteq \mathcal{V}$ such that $|\mathcal{V}(\kappa)| = K_t$ and

$$\sum_{x_t \in \mathcal{V} \setminus \mathcal{V}(\kappa)} p(x_t|\boldsymbol{q}, \boldsymbol{x}_{0:t-1}) \leq \kappa.$$

Then, we have for the truncated approximate power distribution:

$$\hat{p}^{(\text{pow})}_{\alpha,\kappa}(x_t \mid \boldsymbol{q}, \boldsymbol{x}_{0:t-1}) = \frac{p^{\alpha}(x_t \mid \boldsymbol{q}, \boldsymbol{x}_{0:t-1})\, \hat{\zeta}_t(x_t)}{\sum_{x'_t \in \mathcal{V}(\kappa)} p^{\alpha}(x'_t \mid \boldsymbol{q}, \boldsymbol{x}_{0:t-1})\, \hat{\zeta}_t(x'_t)}$$

$$= \frac{p^{\alpha}(x_t \mid \boldsymbol{q}, \boldsymbol{x}_{0:t-1})\, \hat{\zeta}_t(x_t)}{\sum_{x'_t \in \mathcal{V}} p^{\alpha}(x'_t \mid \boldsymbol{q}, \boldsymbol{x}_{0:t-1})\, \hat{\zeta}_t(x'_t) - \kappa}$$

$$= \underbrace{\frac{p^{\alpha}(x_t \mid \boldsymbol{q}, \boldsymbol{x}_{0:t-1})\, \hat{\zeta}_t(x_t)}{\sum_{x'_t \in \mathcal{V}} p^{\alpha}(x'_t \mid \boldsymbol{q}, \boldsymbol{x}_{0:t-1})\, \hat{\zeta}_t(x'_t)}}_{\hat{p}^{(\text{pow})}_{\alpha}(x_t|\boldsymbol{q}, \boldsymbol{x}_{0:t-1})}$$

$$+ \kappa \underbrace{\frac{p^{\alpha}(x_t \mid \boldsymbol{q}, \boldsymbol{x}_{0:t-1})\, \hat{\zeta}_t(x_t)}{\sum_{x'_t \in \mathcal{V}} p^{\alpha}(x'_t \mid \boldsymbol{q}, \boldsymbol{x}_{0:t-1})\, \hat{\zeta}_t(x'_t) \sum_{x'_t \in \mathcal{V}(\kappa)} p^{\alpha}(x'_t \mid \boldsymbol{q}, \boldsymbol{x}_{0:t-1})\, \hat{\zeta}_t(x'_t)}}_{\mathcal{O}(\kappa)}$$

$$= \hat{p}^{(\text{pow})}_{\alpha}(x_t \mid \boldsymbol{q}, \boldsymbol{x}_{0:t-1}) + \mathcal{O}(\kappa).$$

In other words, there is an additional bias term bounded by the parameter $\kappa$. Please notice that, for the sampling of the whole trajectory, this additional bias will accumulate:

$$\prod_{t=0}^{T} \hat{p}_{\alpha,\kappa}^{(\text{pow})}(x_t \mid \boldsymbol{q}, \boldsymbol{x}_{0:t-1}) = \prod_{t=0}^{T} \hat{p}_{\alpha}^{(\text{pow})}(x_t \mid \boldsymbol{q}, \boldsymbol{x}_{0:t-1}) + \mathcal{O}\left( \kappa \sum_{t=0}^{T} \hat{p}_{\alpha}^{(\text{pow})}(x_t \mid \boldsymbol{q}, \boldsymbol{x}_{0:t-1}) \right).$$

Now, to illustrate the effect of horizon truncation $H_t$, let us consider a rollout expression:

$$\hat{\zeta}_t(x_t) = \frac{1}{M_t} \sum_{r=1}^{M_t} p^{\alpha-1}(\boldsymbol{x}_{t+1:T}^{[r]}|\boldsymbol{q}, \boldsymbol{x}_{0:t-1}, x_t)$$

$$= \frac{1}{M_t} \sum_{r=1}^{M_t} p^{\alpha-1}(\boldsymbol{x}_{t+H_t+1:T}^{[r]}|\boldsymbol{q}, \boldsymbol{x}_{0:t-1}, x_t, \boldsymbol{x}_{t+1:t+H_t}^{[r]}) p^{\alpha-1}(\boldsymbol{x}_{t+1:t+H_t}^{[r]}|\boldsymbol{q}, \boldsymbol{x}_{0:t-1}, x_t)$$

$$= \underbrace{\frac{1}{M_t} \sum_{r=1}^{M_t} p^{\alpha-1}(\boldsymbol{x}_{t+1:t+H_t}^{[r]}|\boldsymbol{q}, \boldsymbol{x}_{0:t-1}, x_t)}_{\hat{\zeta}_{t,H_t}(x_t)}$$

$$+ \underbrace{\frac{1}{M_t} \sum_{r=1}^{M_t} \left[ p^{\alpha-1}(\boldsymbol{x}_{t+H_t+1:T}^{[r]}|\boldsymbol{q}, \boldsymbol{x}_{0:t-1}, x_t, \boldsymbol{x}_{t+1:t+H_t}^{[r]}) - 1 \right] p^{\alpha-1}(\boldsymbol{x}_{t+1:t+H_t}^{[r]}|\boldsymbol{q}, \boldsymbol{x}_{0:t-1}, x_t)}_{\eta_{t,H_t}}.$$

Here, $\hat{\zeta}_{t,H_t}(x_t)$ is the truncated term used in Algorithm 2, and $\eta_{t,H_t}$ is a bias introduced by the lookahead parameter $H_t$. Hence:

$$\hat{p}_{\alpha,H_t}^{(\text{pow})}(x_t \mid \boldsymbol{q}, \boldsymbol{x}_{0:t-1}) = \frac{p^{\alpha}(x_t \mid \boldsymbol{q}, \boldsymbol{x}_{0:t-1}) \hat{\zeta}_{t,H_t}(x_t)}{\sum_{x_t' \in \mathcal{V}} p^{\alpha}(x_t' \mid \boldsymbol{q}, \boldsymbol{x}_{0:t-1}) \hat{\zeta}_{t,H_t}(x_t')}$$

$$= \frac{p^{\alpha}(x_t \mid \boldsymbol{q}, \boldsymbol{x}_{0:t-1}) [\hat{\zeta}_t(x_t) - \eta_{t,H_t}]}{\sum_{x_t' \in \mathcal{V}} p^{\alpha}(x_t' \mid \boldsymbol{q}, \boldsymbol{x}_{0:t-1}) [\hat{\zeta}_t(x_t') - \eta_{t,H_t}']}$$

$$= \underbrace{\frac{p^{\alpha}(x_t \mid \boldsymbol{q}, \boldsymbol{x}_{0:t-1}) \hat{\zeta}_t(x_t)}{\sum_{x_t' \in \mathcal{V}} p^{\alpha}(x_t' \mid \boldsymbol{q}, \boldsymbol{x}_{0:t-1}) \hat{\zeta}_t(x_t')}}_{\hat{p}_{\alpha}^{(\text{pow})}(x_t|\boldsymbol{q}, \boldsymbol{x}_{0:t-1})} + \mathcal{O}\left( \sum_{x_t' \in \mathcal{V}} \eta_{t,H_t}' \right).$$

In other words, the lookahead term $H_t$ introduces additional bias given by the sum of parameters $\eta_{t,H_t}'$, where

$$\eta_{t,H_t}' = \frac{1}{M_t} \sum_{r=1}^{M_t} \left[ p^{\alpha-1}(\boldsymbol{x}_{t+H_t+1:T}^{[r]}|\boldsymbol{q}, \boldsymbol{x}_{0:t-1}, x_t', \boldsymbol{x}_{t+1:t+H_t}^{[r]}) - 1 \right] p^{\alpha-1}(\boldsymbol{x}_{t+1:t+H_t}^{[r]}|\boldsymbol{q}, \boldsymbol{x}_{0:t-1}, x_t').$$

To bound this term, we need to quantify the difference

$$p^{\alpha-1}(\boldsymbol{x}_{t+H_t+1:T}^{[r]}|\boldsymbol{q}, \boldsymbol{x}_{0:t-1}, x_t', \boldsymbol{x}_{t+1:t+H_t}^{[r]}) - 1,$$

which specifies how the probability of the tail sequence $\boldsymbol{x}_{t+H_t+1:T}^{[r]}$ depends on the length of the prefix $(\boldsymbol{x}_{0:t-1}, x_t', \boldsymbol{x}_{t+1:t+H_t}^{[r]})$. This can be characterized only by introducing additional assumptions on the model probability $p(\cdot|\boldsymbol{q}, \boldsymbol{x}_{0:t-1})$.

## B.2. Numerical Stability of the Jackknife Estimator

Potentially, the jackknife distributions $\hat{p}_{\alpha,\text{JK}}^{(\text{pow})}(x_t|\boldsymbol{q}, \boldsymbol{x}_{0:t-1})$ can take negative values, although they always sum to 1. We study when such a situation can happen. Let us denote

$$f^{[r]}(x_t) = p^{\alpha}(x_t|\boldsymbol{q}, \boldsymbol{x}_{0:t-1}) p^{\alpha-1}(\boldsymbol{x}_{t+1:T}^{[r]}|\boldsymbol{q}, \boldsymbol{x}_{0:t-1}, x_t) = p(x_t|\boldsymbol{q}, \boldsymbol{x}_{0:t-1}) p^{\alpha-1}(x_t, \boldsymbol{x}_{t+1:T}^{[r]}|\boldsymbol{q}, \boldsymbol{x}_{0:t-1}).$$

Then

$$\hat{p}_{\alpha,\text{JK}}^{(\text{pow})}(x_t|\boldsymbol{q}, \boldsymbol{x}_{0:t-1})$$

$$= M_t \frac{\sum_{r=1}^{M_t} f^{[r]}(x_t)}{\sum_{x_t' \in \mathcal{V}} \sum_{r=1}^{M_t} f^{[r]}(x_t')} - \frac{M_t - 1}{M_t} \sum_{s=1}^{M_t} \frac{\sum_{r \neq s}^{M_t} f^{[r]}(x_t)}{\sum_{x_t' \in \mathcal{V}} \sum_{r \neq s}^{M_t} f^{[r]}(x_t')}$$

$$= \sum_{s=1}^{M_t} \frac{\sum_{r=1}^{M_t} f^{[r]}(x_t)}{\sum_{x_t' \in \mathcal{V}} \sum_{r=1}^{M_t} f^{[r]}(x_t')} - \sum_{s=1}^{M_t} \frac{M_t - 1}{M_t} \frac{\sum_{r \neq s}^{M_t} f^{[r]}(x_t)}{\sum_{x_t' \in \mathcal{V}} \sum_{r \neq s}^{M_t} f^{[r]}(x_t')}$$

$$= \sum_{s=1}^{M_t} \frac{M_t \sum_{r=1}^{M_t} f^{[r]}(x_t) \sum_{x_t' \in \mathcal{V}} \sum_{r \neq s}^{M_t} f^{[r]}(x_t') - (M_t - 1) \sum_{r \neq s}^{M_t} f^{[r]}(x_t) \sum_{x_t' \in \mathcal{V}} \sum_{r=1}^{M_t} f^{[r]}(x_t')}{M_t \sum_{x_t' \in \mathcal{V}} \sum_{r \neq s}^{M_t} f^{[r]}(x_t') \sum_{x_t' \in \mathcal{V}} \sum_{r=1}^{M_t} f^{[r]}(x_t')}$$

$$= \sum_{s=1}^{M_t} \frac{\sum_{r=1}^{M_t} f^{[r]}(x_t) \sum_{x_t' \in \mathcal{V}} \sum_{r=1}^{M_t} f^{[r]}(x_t') - M_t \sum_{r=1}^{M_t} f^{[r]}(x_t) \sum_{x_t' \in \mathcal{V}} f^{[s]}(x_t') + (M_t - 1) f^{[s]}(x_t) \sum_{x_t' \in \mathcal{V}} \sum_{r=1}^{M_t} f^{[r]}(x_t')}{M_t \sum_{x_t' \in \mathcal{V}} \sum_{r \neq s}^{M_t} f^{[r]}(x_t') \sum_{x_t' \in \mathcal{V}} \sum_{r=1}^{M_t} f^{[r]}(x_t')}.$$

Let us denote

$$\Delta = \sum_{r=1}^{M_t} f^{[r]}(x_t) \sum_{x_t' \in \mathcal{V}} \sum_{r=1}^{M_t} f^{[r]}(x_t')$$

and

$$\Gamma_s = M_t \sum_{x_t' \in \mathcal{V}} \sum_{r \neq s}^{M_t} f^{[r]}(x_t') \sum_{x_t' \in \mathcal{V}} \sum_{r=1}^{M_t} f^{[r]}(x_t').$$

Then we have:

$$\hat{p}_{\alpha,\text{JK}}^{(\text{pow})}(x_t|\boldsymbol{q}, \boldsymbol{x}_{0:t-1}) = \sum_{s=1}^{M_t} \frac{\Delta - M_t \sum_{r=1}^{M_t} f^{[r]}(x_t) \sum_{x_t' \in \mathcal{V}} f^{[s]}(x_t') + (M_t - 1) f^{[s]}(x_t) \sum_{x_t' \in \mathcal{V}} \sum_{r=1}^{M_t} f^{[r]}(x_t')}{\Gamma_s}. \quad (23)$$

In order to make the overall expression negative, we need the term $M_t \sum_{r=1}^{M_t} f^{[r]}(x_t) \sum_{x_t' \in \mathcal{V}} f^{[s]}(x_t')$ to strongly dominate the term $(M_t - 1) f^{[s]}(x_t) \sum_{x_t' \in \mathcal{V}} \sum_{r=1}^{M_t} f^{[r]}(x_t')$ where we even ignore the positive term $\Delta$. Let us investigate what this strong dominance really implies:

$$M_t \sum_{r=1}^{M_t} f^{[r]}(x_t) \sum_{x_t' \in \mathcal{V}} f^{[s]}(x_t') \gg (M_t - 1) f^{[s]}(x_t) \sum_{x_t' \in \mathcal{V}} \sum_{r=1}^{M_t} f^{[r]}(x_t') \implies$$

$$M_t \sum_{r=1}^{M_t} f^{[r]}(x_t) \left[ f^{[s]}(x_t) + \sum_{x_t' \neq x_t} f^{[s]}(x_t') \right] \gg (M_t - 1) f^{[s]}(x_t) \left[ \sum_{r=1}^{M_t} f^{[r]}(x_t) + \sum_{x_t' \neq x_t} \sum_{r=1}^{M_t} f^{[r]}(x_t') \right] \implies$$

$$M_t \left[ f^{[s]}(x_t) + \sum_{r \neq s} f^{[r]}(x_t) \right] \sum_{x_t' \neq x_t} f^{[s]}(x_t') \gg (M_t - 1) f^{[s]}(x_t) \sum_{x_t' \neq x_t} \left[ \sum_{r \neq s} f^{[r]}(x_t') + f^{[s]}(x_t') \right] \implies$$

$$M_t \sum_{r \neq s} f^{[r]}(x_t) \sum_{x_t' \neq x_t} f^{[s]}(x_t') \gg (M_t - 1) f^{[s]}(x_t) \sum_{x_t' \neq x_t} \sum_{r \neq s} f^{[r]}(x_t').$$

Notice that the above relation will be valid if:

1. For token $x_t$, the function $f^{[s]}(x_t)$ is much less than the functions $f^{[r]}(x_t)$ for $r \in \{1, \ldots, M_t\} \setminus \{s\}$.

2. For other tokens $x_t' \in \mathcal{V} \setminus \{x_t\}$, the functions $f^{[r]}(x_t')$ are much less than the function $f^{[s]}(x_t')$ for all $r \in \{1, \ldots, M_t\} \setminus \{s\}$.

In other words, this implies that for token $x_t$ we have a rollout $\boldsymbol{x}_{t+1:T}^{[s]} \sim p(\cdot|\boldsymbol{q}, \boldsymbol{x}_{0:t-1}, x_t)$ with a much smaller value

$$p(x_t|\boldsymbol{q}, \boldsymbol{x}_{0:t-1}) p^{\alpha-1}(x_t, \boldsymbol{x}_{t+1:T}^{[s]}|\boldsymbol{q}, \boldsymbol{x}_{0:t-1})$$

| Task | Qwen Prompt | DeepSeek Prompt |
|---|---|---|
| **MATH500** | `Can you solve the following math problem?`
`Please reason step by step, and put your final answer within \\boxed{}` | `Please reason step by step, and put your final answer within \\boxed{}` |
| **HumanEval** | `Write a Python function to solve the following problem:` | `Complete the following Python function:` |
| **GPQA** | `Answer the following multiple-choice question. The last line of your response should be of the following format: '\\boxed{LETTER}' (without quotes) where LETTER is one of ABCD (ex. '\\boxed{A}'). Think step by step before answering.`
`{Question} A) {} B) {} C) {} D) {}` | `Answer the following multiple-choice question. The last line of your response should be of the following format: '\\boxed{LETTER}' (without quotes) where LETTER is one of ABCD (ex. '\\boxed{A}'). Think step by step before answering.`
`{Question} A) {} B) {} C) {} D) {}` |

*Table 4.* Prompt templates used for baselines and our method. Adapted from the MCMC power-sampling repository (Karan & Du, 2025) and DeepSeek official instruction format.

than the values

$$p(x_t|\boldsymbol{q}, \boldsymbol{x}_{0:t-1})p^{\alpha-1}(x_t, \boldsymbol{x}_{t+1:T}^{[r]}|\boldsymbol{q}, \boldsymbol{x}_{0:t-1})$$

for $r \neq s$. Additionally, the second condition implies that for all other tokens $x'_t \neq x_t$, there is a rollout $\boldsymbol{x}'^{[s]}_{t+1:T} \sim p(\cdot|\boldsymbol{q}, \boldsymbol{x}_{0:t-1}, x'_t)$ with a much larger value

$$p(x'_t|\boldsymbol{q}, \boldsymbol{x}_{0:t-1})p^{\alpha-1}(x'_t, \boldsymbol{x}'^{[s]}_{t+1:T}|\boldsymbol{q}, \boldsymbol{x}_{0:t-1})$$

than the values

$$p(x'_t|\boldsymbol{q}, \boldsymbol{x}_{0:t-1})p^{\alpha-1}(x'_t, \boldsymbol{x}'^{[r]}_{t+1:T}|\boldsymbol{q}, \boldsymbol{x}_{0:t-1})$$

for $r \neq s$. In practice, this situation is rare for LLM generations, since it requires a single rollout to be an extreme outlier for one token while simultaneously dominating the corresponding rollout statistics for the other candidate tokens. We did not observe negative jackknife probabilities in our experiments. Both such cases can be fixed by resampling the rollout $\boldsymbol{x}^{[s]}_{t+1:T}$ for token $x_t$ and the rollouts $\boldsymbol{x}'^{[s]}_{t+1:T}$ for other tokens $x'_t \neq x_t$.

# C. Further Experimental Results

## C.1. Experimental Setups and Implementation Details

We implement our method following Algorithm 2. All experiments are run on the same single GPU using `vLLM` v0.6.3. Since our approach is purely a sampling procedure (no backpropagation). Therefore, compared to post-training approaches, the method is comparatively lightweight in memory and can be run on small-memory devices.

**Prompts.** Table 4 lists the prompt templates used for each benchmark. For the Qwen models, we adopt the prompt templates used in the MCMC power-sampling repository to ensure fairness. For the DeepSeek models, we follow the official DeepSeek instruction format, keeping the task content identical while changing only the wrapper (instruction formatting). All the methods reported in the main evaluation section use these prompts.

**Hyperparameter settings.** We summarise the key hyperparameters used for each sampling method in Table 5. Across all methods, we use a maximum generation length of $T_{\max} = 3072$ tokens and terminate early upon emitting an EOS token. Unless otherwise stated, we set the random seed to 0 for all main results. For evaluation, we use the same scoring functions as the open-source MCMC power-sampling implementation (Karan & Du, 2025) to ensure a fair comparison across decoding strategies.

| Method | Hyperparameters |
|---|---|
| Low-temperature | temperature $= 0.25 = 1/\alpha$ (with $\alpha = 4$) |
| Best-of-$N$ | $N_{\text{BoN}} = 32$; selection score $\log p(\boldsymbol{x} \mid \boldsymbol{q})$ |
| MCMC power sampling | $\alpha = 4$, $B = 192$, $N_{\text{MCMC}} = 10$, $p_{\text{prop}}$: base temperature $1/\alpha$ |
| Power Sampling (ours) | $\alpha = 4$, $B = 192$, $K_t = 8$, $M_t = 8$ |

*Table 5.* Main hyperparameter settings for each method.

For Low-temperature sampling, we apply token-level temperature scaling with temperature $1/\alpha$ and set $\alpha = 4$.

For the verifier-free version of Best-of-$N$, we generate $N_{\text{BoN}} = 32$ independent completions and select the candidate with the highest base-model sequence score, ranked by $\log p(\boldsymbol{x} \mid \boldsymbol{q})$ (using the same normalisation choice across benchmarks).

For the MCMC sampling method, we follow the open-source implementation of (Karan & Du, 2025) with $\alpha = 4$, $T_{\max} = 3072$, block size $B = 192$, $N_{\text{MCMC}} = 10$ MCMC steps per stage, and proposal model $p_{\text{prop}}$ given by the base model sampled at temperature $1/\alpha = 0.25$.

For the GRPO baseline, we use the GRPO checkpoints from (Shao et al., 2025; 2024). Training hyperparameters are described in the corresponding papers, and the released model weights are available at [2] and [3]. These checkpoints correspond to post-training on the MATH training split for Qwen2.5-Math-7B and DeepSeek-Math-7B, respectively.

Our autoregressive approximation uses the same global settings as MCMC power sampling ($\alpha = 4$, $T_{\max} = 3072$, $B = 192$) and applies token-level lookahead with Top-$K_t$ candidates and $M_t$ rollouts per candidate; we set $K_t = M_t = 8$ and use the jackknife-corrected estimator $\hat{p}^{(\text{pow})}_{\alpha, \text{JK}}$ throughout. All likelihood computations are performed in log-space for numerical stability.

**Inference-time measurement and practical scaling.** All reported inference times are measured on a single GPU under the same runtime stack. We report wall-clock *generation time per prompt* (excluding dataset loading and metric computation), averaged over the full evaluation set with the same batching configuration across methods, and fix the random seed to ensure comparable decoding behaviour. In our current configuration, our method is typically about 2.5–3.5$\times$ slower than standard decoding due to additional rollouts, but remains up to 10$\times$ faster than MCMC power sampling, which requires iterative refinement steps. The overhead of our method scales primarily with the rollout budget and candidate set size $(K_t, M_t)$ and the maximum generation length $T_{\max}$, and can be reduced by tuning these budgets with limited accuracy loss (see ablation study). Compared with RL post-training methods, our method requires no training compute or external rewards, while post-training can improve pass@1, it typically requires many generations per training example over large training sets, whereas sampling-based sharpening can be deployed immediately and is particularly attractive in small-to-moderate deployment regimes.

### C.2. More Pass@k Results and Analysis

**Pass@K on out-of-domain tasks.** Figure 5 reports pass@k on HumanEval and GPQA for Qwen2.5-Math-7B and its GRPO (MATH) post-trained variant. Although both tasks are out-of-domain with respect to GRPO training (which uses the MATH training split), we observe qualitatively different behaviours across domains. On HumanEval, GRPO retains comparable pass@k scaling to the base model, suggesting no strong diversity collapse in this setting. In contrast, on GPQA the GRPO curve shows similar behaviour as the in-domain MATH500 task (in Figure 3): GRPO boosts pass@1 but provides limited additional gains for larger $k$, consistent with diversity reduction, suggesting that this kind of collapse also persists even in the out-of-domain task of GRPO. This behaviour is broadly consistent with recent findings that reward-based post-training can improve pass@1 while reducing diversity at larger $k$ (Yue et al., 2025), and we leave a systematic multi-model study of this effect on different post-training models to future work.

In comparison, power sampling methods improve pass@1 without inducing the same degradation in pass@k scaling. Both MCMC power sampling and our method achieve strong pass@k growth. This supports the main message of the paper:

---

[2] https://huggingface.co/stellalisy/rethink_rlvr_reproduce-ground-truth-qwen2.5_math_7b-lr5e-7-kl0.00-step150

[3] https://huggingface.co/deepseek-ai/deepseek-math-7b-rl

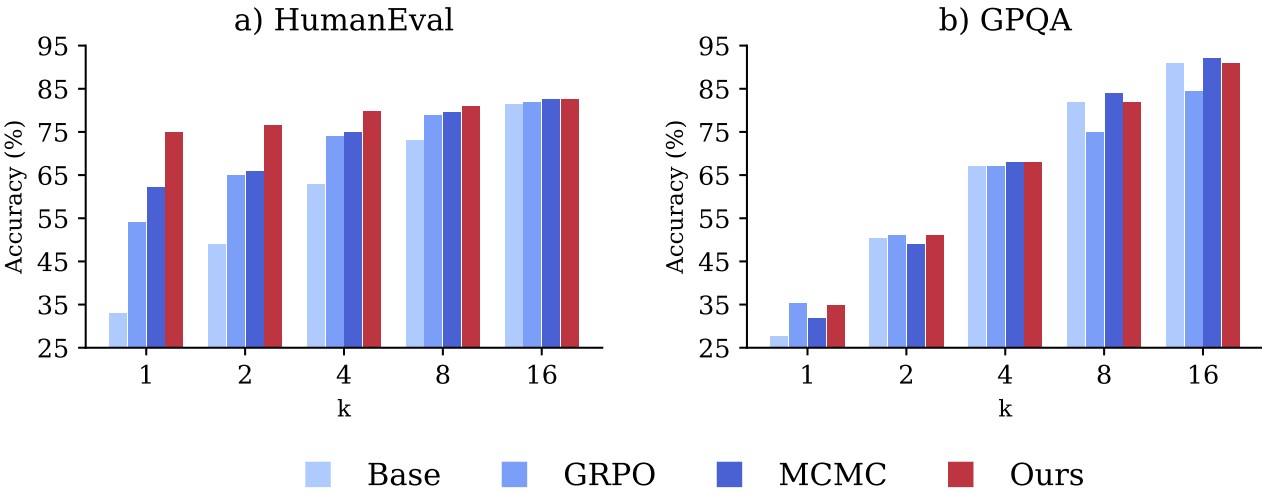

*Figure 5.* Pass@k performance $k \in \{1, 2, 4, 8, 16\}$ on out-of-domain tasks ((a) HumanEval and (b) GPQA). On HumanEval, GRPO exhibits similar scaling with $k$ to the base model (no obvious diversity collapse). On GPQA, GRPO improves pass@1 but scales more weakly with $k$, indicating reduced diversity even on an out-of-domain benchmark. In contrast, power sampling (MCMC and ours) improves pass@1 while preserving strong pass@k scaling.

inference-time distribution sharpening can increase the probability of sampling high-quality trajectories while preserving diversity, without requiring training or external verifiers.

**Qualitative examples.** We include two representative failure cases for GRPO under the pass@k setting, one in-domain example from MATH500 and one out-of-domain example from GPQA, that remain incorrect at pass@16 (as shown in Tables 6 and 7, we abbreviate long generations with "..." for readability). We observe that these failures are often driven less by a lack of capability than by reduced diversity: across 16 samples in practice, the GRPO model frequently repeats near-identical reasoning and the same final answer on many problems, so increasing $k$ provides limited additional coverage. This behaviour is not observed in non-RL models evaluated under identical prompts and sampling parameters, including Qwen2.5-Math-7B and DeepSeek-Math-7B, which are also trained on math-related data and typically produce a broader set of distinct candidates, thereby benefiting more from pass@k.

The GPQA example further illustrates a common out-of-domain failure mode: in the QA task, the GRPO-MATH model can produce largely correct reasoning yet still answer incorrectly by selecting the wrong option letter, or by terminating after intermediate calculations without outputting a final multiple-choice selection. This failure mode is less frequent in the corresponding base model, suggesting it may be a side effect of GRPO post-training on the MATH training split for this checkpoint. In contrast, our power sampling recovers correct trajectories by exploring alternative solution paths while preserving stochastic diversity.

### C.3. Additional Ablation Studies

**Hyperparameter sensitivity across models.** We conduct additional ablations to assess sensitivity to hyperparameters on Qwen2.5-7B and DeepSeek-Math-7B in math, code, QA tasks separately on a validation subset. Figure 6 shows consistent trends across models and tasks, which align with our findings in Qwen2.5-Math-7B (in Figure 4). First, performance is robust to the rollout and candidate budgets: under the same sharpening exponent $\alpha$, varying $M_t = K_t$ over the tested range only mildly affects pass@1, indicating that most gains can be achieved with relatively small sampling budgets. Second, for all the three base models, the strongest region is typically in a moderate range ($\alpha \approx 4$–5), while overly aggressive sharpening (e.g., $\alpha = 8$) can degrade performance, and $\alpha = 1$ performs poorly on most tasks. A notable exception appears on GPQA for DeepSeek-Math-7B, $\alpha = 1$ can be competitive in parts of the budget range, which also aligns with the empirical observation that low-temperature decoding can be particularly strong on GPQA for this model. Overall, these ablations suggest that power sampling can "unlock" improved pass@1 across diverse base models without post-training using a small and stable hyperparameter range, and that the optimal $\alpha$ reflects a task-dependent balance between sharpening

---

**Example 2 (Task 82, MATH500, Qwen2.5-Math-7B).**

---

**Question.** Point $A$ lies somewhere within or on the square with opposite corners $(0,0)$ and $(2,2)$. Point $B$ lies somewhere within or on the square with opposite corners $(4,2)$ and $(5,3)$. What is the greatest possible value of the slope of the line containing points $A$ and $B$? Express your answer as a common fraction.

**GRPO (MATH).**

To determine the greatest possible value of the slope of the line containing points $A$ and $B$, we need to consider the coordinates of the points $A$ and $B$ within their respective squares. The slope of a line passing through points $(x_1, y_1)$ and $(x_2, y_2)$ is given by $\frac{y_2-y_1}{x_2-x_1}$. To maximize this slope, we need to maximize the difference in the $y$-coordinates and minimize the difference in the $x$-coordinates. ...

The coordinates of point $A$ can range from $(0,0)$ to $(2,2)$, and the coordinates of point $B$ can range from $(4,2)$ to $(5,3)$. ... This means we should consider the points $(0,0)$ for $A$ and $(4,3)$ for $B$, as these points will give the maximum possible slope. ...

Let's calculate the slope for these points:
$$\text{slope} = \frac{3-0}{4-0} = \frac{3}{4}.$$

Let's verify this using Python code to ensure accuracy. ... `output 3.0`
Therefore, the greatest possible value of the slope of the line containing points $A$ and $B$ is:

$$\boxed{\frac{3}{4}}.$$

---

**Ours.**

To determine the greatest possible value of the slope of the line containing points $A$ and $B$, we need to consider the coordinates of the points $A$ and $B$ within their respective squares. The slope $m$ of the line passing through points $A(x_1, y_1)$ and $B(x_2, y_2)$ is given by:
$$m = \frac{y_2 - y_1}{x_2 - x_1}.$$

The coordinates of point $A$ can range from $(0,0)$ to $(2,2)$, and the coordinates of point $B$ can range from $(4,2)$ to $(5,3)$. To maximize the slope, we need to maximize the numerator $y_2 - y_1$ and minimize the denominator $x_2 - x_1$. ...

The maximum value of $y_2 - y_1$ occurs when $y_2 = 3$ and $y_1 = 0$, giving $y_2 - y_1 = 3$. The minimum value of $x_2 - x_1$ occurs when $x_2 = 4$ and $x_1 = 2$, giving $x_2 - x_1 = 2$. Therefore, the greatest possible value of the slope is:

$$m = \frac{3}{2}.$$

Let's confirm this with Python code ... `output 1.5`
Thus, the final answer is:

$$\boxed{\frac{3}{2}}.$$

---

**Ground truth.** $\boxed{\dfrac{3}{2}}$

---

*Table 6.* Qualitative example illustrating pass@k behaviour. We show one sampled completion from GRPO (MATH) and our method on the same prompt.

---

**Example 2 (Task 19, GPQA-Diamond, Qwen2.5-Math-7B).**

---

**Question.** A common approximation made in many-body nuclear calculations of some interaction (for example an electroweak interaction) is to suppose that nucleons in a nucleus behave in the same way as free nucleons. Which of the following four assumptions jointly imply this "impulse" approximation?
1. The interaction current only interacts with individual nucleons.
2. The nucleus is transparent apart from the selected nucleon.
3. The quarks internal to the selected nucleon are non-relativistic.
4. The interaction proceeds as if the selected nucleon experiences no binding forces.
A) 2,3,4   B) 1,2,4   C) 1,2,3   D) 1,3,4

---

**GRPO (MATH).**

To answer this question, we need to understand the conditions under which the impulse approximation holds. The impulse approximation is a simplification used in many-body nuclear calculations where the interaction current only interacts with individual nucleons, the nucleus is transparent apart from the selected nucleon, . . . and the interaction proceeds as if the selected nucleon experiences no binding forces.
Let's break down the conditions: 1. The interaction current only interacts with individual nucleons.
2. The nucleus is transparent apart from the selected nucleon.
3. The quarks internal to the selected nucleon are non-relativistic.
4. The interaction proceeds as if the selected nucleon experiences no binding forces.
The impulse approximation is valid when these conditions are met. Therefore, the correct answer is the set of conditions that includes all of these. . . .
Let's check the options: A) 2,3,4    B) 1,3,4    C) 1,2,4    D) 1,2,3
The correct answer is C) 1,2,4 because it includes all the conditions required for the impulse approximation. . . .

$$\boxed{C}.$$

---

**Ours.**
To determine which assumptions jointly imply the "impulse" approximation, let's analyze each assumption step by step:
1. The interaction current only interacts with individual nucleons. . . . This is a key aspect of the impulse approximation. . . .
2. The nucleus is transparent apart from the selected nucleon. . . . This is another important aspect of the impulse approximation. . . .
3. The quarks internal to the selected nucleon are non-relativistic. . . . not directly related to the impulse approximation. . . .
4. The interaction proceeds as if the selected nucleon experiences no binding forces. . . . another key aspect of the impulse approximation. . . .
Based on this analysis, the assumptions that jointly imply the impulse approximation are 1, 2, and 4. Therefore, the correct answer is:

$$\boxed{B}.$$

---

**Ground truth.** $\boxed{B}$

---

*Table 7.* Qualitative example on GPQA illustrating pass@k behaviour. We show one sampled completion from GRPO (MATH) and our method on the same prompt.

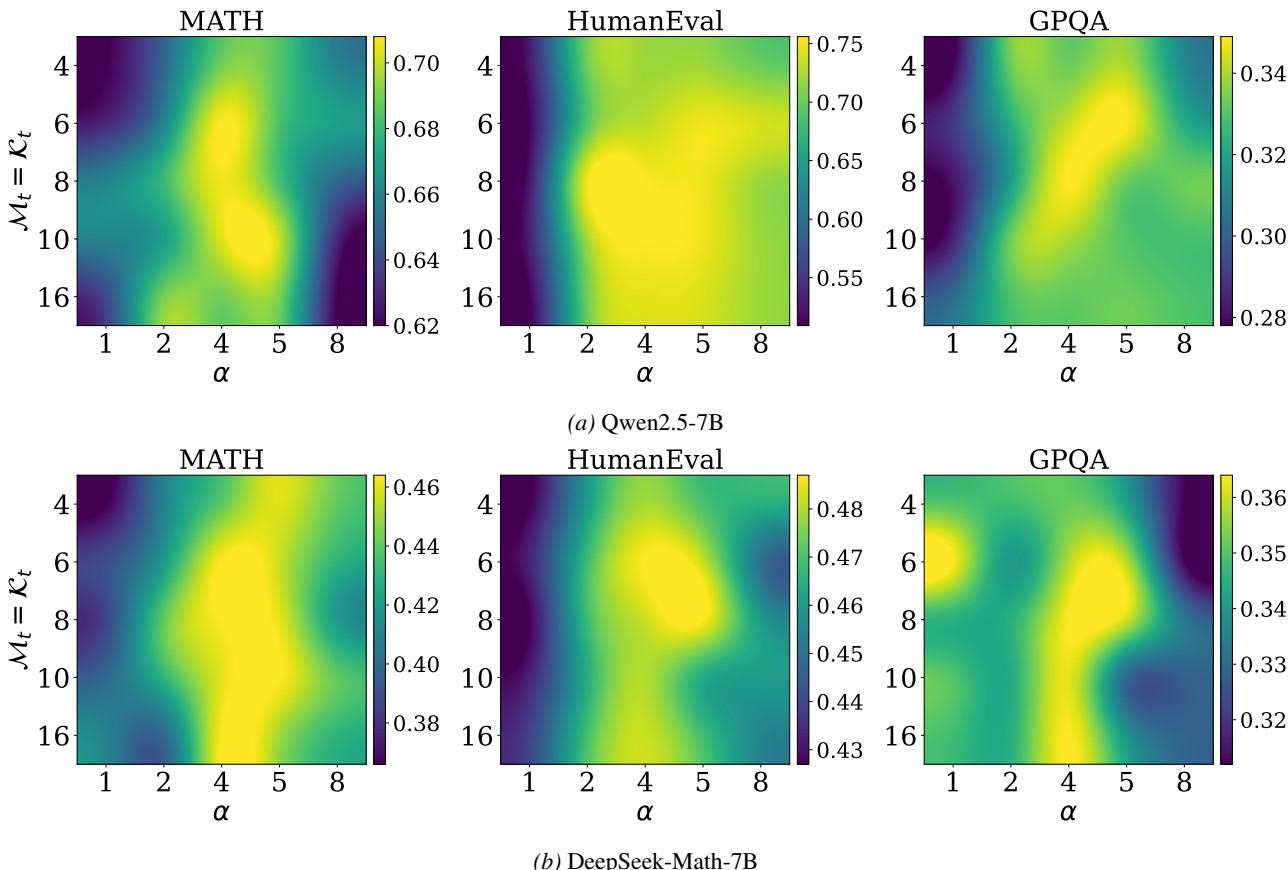

*(a)* Qwen2.5-7B

*(b)* DeepSeek-Math-7B

*Figure 6.* Pass@1 performance for MATH, HumanEval, and GPQA with $\alpha \in \{1, 2, 4, 5, 8\}$ (x-axis) and $M_t = K_t \in \{4, 6, 8, 10, 16\}$ (y-axis) in Qwen2.5-7B (top) and DeepSeek-Math-7B (bottom).

and preserving exploratory diversity.

| $H_t$ | Acc. (%) | Time (min/prompt) |
|-------|----------|-------------------|
| 48    | 74.8     | 0.09              |
| 96    | 75.6     | 0.16              |
| 192   | 75.8     | 0.22              |
| 384   | 76.2     | 0.29              |
| 768   | 74.8     | 0.38              |

*(a) Varying $H_t$ with $B = 192$.*

| $B$ | Acc. (%) | Time (min/prompt) |
|-----|----------|-------------------|
| 48  | 76.4     | 0.43              |
| 96  | 74.8     | 0.26              |
| 192 | 75.8     | 0.22              |
| 384 | 74.8     | 0.14              |
| 768 | 70.4     | 0.11              |

*(b) Varying $B$ with $H_t = 192$.*

*Table 8.* Effect of truncated lookahead horizon $H_t$ and block size $B$ on MATH500 with Qwen2.5-Math-7B.

Next, we study two parameters of the batched algorithm: the truncated lookahead horizon $H_t$ and the block size $B$. The horizon $H_t$ controls how far each rollout looks into the future, while $B$ controls how many tokens are generated before recomputing the power-sampling correction. As shown in Table 8, increasing $H_t$ generally increases latency because each rollout extends further into the future. The performance improves up to a moderate range but is not strictly monotonic, reflecting the stochastic nature of rollout-based estimation and the fact that long lookahead is not always necessary. For $B$, larger block sizes reduce latency by applying the correction less frequently, but overly large batches can degrade accuracy because fewer future-aware corrections are applied during generation. In our main results, we use $H_t = 192$ and $B = 192$, matching the baseline setting while providing a practical accuracy–latency trade-off.

**Jackknife bias correction.** We next ablate the jackknife correction used in our estimator. The correction is designed to reduce the finite-sample bias that arises when Monte Carlo estimates are substituted into a ratio estimator. As shown in Table 9, the naive estimator remains competitive in some settings, but overall removing the jackknife correction leads to an average drop of 4.29% across three base models and three benchmarks. This confirms that bias correction is an important component of the proposed approximation, especially when the rollout budget is limited.

|                                    | MATH500 | HumanEval | GPQA  |
|------------------------------------|---------|-----------|-------|
| **Qwen2.5-7B**                     |         |           |       |
| Base                               | 0.498   | 0.329     | 0.278 |
| Power Sampling (naive estimator)   | 0.650   | 0.677     | 0.318 |
| **Power Sampling (ours)**          | **0.708** | **0.756** | **0.349** |
| **Qwen2.5-Math-7B**                |         |           |       |
| Base                               | 0.496   | 0.329     | 0.278 |
| Power Sampling (naive estimator)   | 0.744   | 0.530     | 0.353 |
| **Power Sampling (ours)**          | **0.758** | **0.604** | **0.409** |
| **DeepSeek-Math-7B**               |         |           |       |
| Base                               | 0.362   | 0.415     | 0.333 |
| Power Sampling (naive estimator)   | 0.416   | **0.487** | 0.338 |
| **Power Sampling (ours)**          | **0.464** | **0.487** | **0.364** |

*Table 9.* Effect of jackknife bias correction. Power Sampling (naive estimator) replaces the jackknife-corrected estimator with the naive rollout estimator, while Power Sampling (ours) denotes our full method.

