# OpenReview forum: "Scalable Power Sampling: Unlocking Efficient, Training-Free Reasoning for LLMs via Distribution Sharpening"
_ICML.cc/2026/Conference — ICML 2026 regular_

### Official Review · Reviewer_HGw3 · 2026-03-05

**Soundness:** 3
**Presentation:** 3
**Significance:** 3
**Originality:** 3
**Overall Recommendation:** 5
**Confidence:** 4

**Summary:**

The authors introduce a reward-free and training-free method for inference-time distribution modification/sharpening, which re-weights token probabilities using estimated future, sequence-level likelihood. Although similar approaches exist, the main contribution of this work is a sampling method that requires fewer samples and therefore is substantially faster than existing approaches (in some cases 10x faster). The proposed approach outperforms competing methods on downstream reasoning benchmarks, and in some cases even outperforms tuning with GRPO.

**Compliance With Llm Reviewing Policy:**

Affirmed.

**Final Justification:**

The authors addressed my concerns: I maintain my already-high scores.

**Key Questions For Authors:**

- What are the scalar factors hidden in the big-O notation ($O(1/M_t)$ and $O(1/M_t^2)$) in Lemmas 3.2 and 3.3?

- The authors state that only the top $K_t$ tokens at each timestep $t$ are re-weighted (L203-204), where $K_t$ is a dynamic value (L252-257). How is $K_t$ calculated?

- How does the inference time for the proposed approach compare to the base model? Similarly, how does inference time compare to GRPO-tuned models? Obviously, GRPO will have the same per-token inference time as the base model, but it may generate longer answers when reasoning, slowing down sequence-level generation times.

- Given that RL training has been shown to mostly sharpen the model's distribution (L020-024), do we see a significant difference in *how* RL sharpens the distribution in comparison to training-free approaches such as the proposed method?

**Typos:**

- "Algorithm" => "Algorithm 2" (L198)

**Limitations:**

No discussion of limitations in the paper. There are two main limitations: the relatively small size of the models used in the experiments (7B), and the fact that the proposed approach still requires sampling multiple completions per token to estimate probabilities (which is very inefficient).

**Strengths And Weaknesses:**

**Strengths:**

- The findings of this paper are interesting, especially the fact that models tuned with GRPO can be further improved using the proposed method, without requiring additional training.

- The proposed method is well-motivated theoretically, and its effectiveness is demonstrated empirically.

- Although this work primarily combines existing techniques, it does so in a way that is novel (to the best of my knowledge).

**Weaknesses:**

- There are a few presentational issues. Proofs are located in the appendix, but are not referenced under the Theorems/Lemmas in the main body. Some important components of the proposed approach are located in the appendix, and must be at least briefly discussed in the main body: e.g. the rollout length truncation factor $H_t$ (L206-207) and batching algorithm (L396).

---

> ### Author Rebuttal · Authors · 2026-03-31
>
> We thank the reviewer for the positive feedback and valuable suggestions for improving this work.
>
> > **Proofs and some important components of the proposed approach are located in the appendix**
>
> We will add a proof sketch in the main body and briefly introduce the batch algorithm and lookahead truncation there as well, with references to the appendix for the full details.
>
> > **What are the scalar factors hidden in the big-O notation ($O(1/M_t)$ and $O(1/M_t^2)$) in Lemmas 3.2 and 3.3?**
>
> The hidden factors can be made explicit as worst-case upper bounds $\mathbb{E}\left[\hat{p}^{(\mathrm{pow})}\_{\alpha}(x\_t \mid \mathbf{q}, \mathbf{x}\_{0:t-1})\right] - p^{(\mathrm{pow})}\_{\alpha}(x\_t \mid \mathbf{q}, \mathbf{x}\_{0:t-1}) \le \frac{4w^2\left(1+|\mathcal{V}|w^2\right)}{M\_t}$ and $\mathbb{E}\left[\hat{p}^{(\mathrm{pow})}\_{\alpha,\mathrm{JK}}(x\_t \mid \mathbf{q}, \mathbf{x}\_{0:t-1})\right] - p^{(\mathrm{pow})}\_{\alpha}(x\_t \mid \mathbf{q}, \mathbf{x}\_{0:t-1}) \le \frac{4w^2\left(12w^2|\mathcal{V}|^2 - 9w^2|\mathcal{V}| - 2w|\mathcal{V}| + 2 - 3w\right)}{M\_t^2}$, where $w = |\mathcal{V}|^{(T-t+1)(\alpha-1)}$. We note that this analysis corresponds to a worst-case scenario, where all possible rollouts $\mathbf{x}_{t:T}$ of length $T-t+1$ are assumed to be equally likely under the LLM. This assumption is generally unrealistic in pretrained LLMs, and the resulting constants can be quite loose in practice.
>
> > **How is $K_t$ calculated?**
>
> We suggest one intuitive way to select $K_t$ dynamically from the next-token distribution at step t, although in practice it could also be fixed or computed in other ways. We can sort tokens by $p(x_t\mid q,x_{0:t-1})$ and retain only those whose probability is above a threshold relative to the maximum next-token probability, i.e., $p(x_t\mid q,x_{0:t-1}) \ge \delta \max_{x'} p(x'\mid q,x_{0:t-1})$. The number of retained tokens then defines $K_t$. Thus, $K_t$ becomes smaller when the distribution is sharp and larger when the distribution is flatter.
>
> > **Inference time compared to the base model and GRPO-tuned model**
>
> We will detail the inference time compared to the base model and GRPO models in the revised paper. On MATH500 with Qwen2.5-Math-7B, the per-prompt time is 0.06 minutes for the base model and 0.22 minutes for our method. The comparison with GRPO is similar to that with the base model, since the average final output lengths are similar between the base model and GRPO model (640 tokens for the base model and 682 for the GRPO model). Although GRPO does produce longer responses on specific questions, its average output length is not very different from that of the base model in this setting. The practical compute trade-off between GRPO and our method also depends on the deployment setting, including the training setup and the number of inference calls. GRPO incurs additional training cost, whereas our method shifts extra computation to inference time.
>
> > **Do we see a significant difference in how RL sharpens the distribution in comparison to training-free approaches?**
>
> Yes, we observe a clear difference in both diversity and concentration. For Qwen2.5-Math-7B, we measured the answer-duplicate rate across 8 rollouts per prompt over 100 samples. The duplicate rate of answers is 58.3\% for low-temperature decoding, 83.3\% for the GRPO model, and 62.5\% for our method. These results suggest that the diversity induced by GRPO and by training-free approaches is quite different. Also, the average mean log-probability per token is -0.12 for low-temperature decoding, -0.10 for our method, and -0.02 for GRPO, indicating that GRPO produces a much sharper and more concentrated distribution. Qualitatively, the GRPO checkpoint of Qwen2.5-Math-7B often follows highly similar reasoning patterns, while the base model and our method remain more diverse. This is also consistent with the weaker pass@K scaling we observe for GRPO.
>
> > **Relative small size of the model used in experiments**
>
> We agree that evaluation on a larger model would further strengthen our claim. We therefore added results on Qwen2.5-32B: our method still provides consistent gains of 20.8\% over the base model while remaining 9.4\% better than MCMC. Our method also remains faster, requiring 0.36 minutes per prompt, compared with 4.1 minutes for MCMC.
>
> | Method | MATH500 | HumanEval | GPQA |
> |---|---:|---:|---:|
> | **Qwen2.5-32B** |  |  |  |
> | Base | 0.577 | 0.392 | 0.376 |
> | Low-temperature | 0.682 | 0.507 | 0.414 |
> | MCMC Power Sampling | 0.738 | 0.567 | 0.383 |
> | Power Sampling (ours) | **0.748** | **0.793** | **0.429** |
>
> **Table 1:** Evaluation on Qwen2.5-32B.

---

> > ### Author Rebuttal · Reviewer_HGw3 · 2026-04-02
> >
> > Thank you for the clarifications and additional experiments. I believe this addresses my concerns. I only won't be raising my score because it is already quite high.

---

> > > ### Author Response · Authors · 2026-04-04
> > >
> > > Thank you for reading our rebuttal and for your positive feedback and helpful suggestions. We are glad that our clarifications addressed your concerns, and we will incorporate these points into the revised paper.

---

### Official Review · Reviewer_2SZZ · 2026-03-07

**Soundness:** 3
**Presentation:** 3
**Significance:** 3
**Originality:** 3
**Overall Recommendation:** 5
**Confidence:** 2

**Summary:**

This paper studies inference-time reasoning and asks whether post-RL gains mainly come from distribution sharpening. The authors derive a decomposition showing that the global power distribution can be written as a low-temperature distribution plus a future-quality correction term. Based on this result, they propose a training-free and verifier-free approximation using autoregressive Monte Carlo lookahead with jackknife bias correction. On four 7B models and three benchmarks, the method reports performance comparable to or better than MCMC power sampling at much lower latency.

**Compliance With Llm Reviewing Policy:**

Affirmed.

**Final Justification:**

This is an interesting work, and I recommend acceptance.

**Key Questions For Authors:**

(1) The performance gain on the post-trained model is minimal, as shown in Tables 1 and 2. Compared to the improvement on the base model, the method's gain on the post-trained model is small, which raises concerns about its practical applicability.

(2) At least one larger-scale model result is needed to support the scalability claim. The authors are expected to report results on a larger model and compare trend consistency with 7B findings.

(3) A proof of bias at the token level has been provided, but extending this to the Batch Tokens Extension lacks a bound proof of bias accumulation.

I will change my score after discussing with the other reviewers and rebuttal.

**Limitations:**

Yes

**Strengths And Weaknesses:**

Strengths:

(1) Empirical results show strong speed-accuracy tradeoffs versus MCMC baselines.
(2) The new perspective presented in the article is very innovative and interesting.
(3) The article has a clear organizational structure.

Weaknesses:

Evidence is still insufficient on estimator stability, statistical reliability, and scalability breadth.

---

> ### Author Rebuttal · Authors · 2026-03-31
>
> We thank the reviewer for the thoughtful questions and positive review.
>
> > **The performance gain on the post-trained model is minimal compared to the improvement on the base model.**
>
> We agree that the gain on the GRPO-tuned model is smaller than on other base models, but we view this as expected rather than as a limitation of practical applicability. The GRPO checkpoint already sharpens the output distribution substantially. For example, on MATH500, the GRPO model often produces highly similar responses (avg. 83.33\% duplicate answers over 8 rollouts), leaving limited room for further gains not only for our method, but also for other sampling-based approaches such as BoN, MCMC, and self-consistency. This is also consistent with our pass@K results: GRPO improves pass@1, but often yields much smaller gains at larger K, indicating reduced sample diversity. Our main practical point is that we recover substantial gains over the base models without any additional training, reward model, or verifier, while still obtaining modest positive gains on top of the GRPO model. The practical gain depends on how much diversity remains after post-training: when post-training already strongly collapses the output distribution, the sampling-based improvement is naturally limited.
>
> > **At least one larger-scale model result is needed to support the scalability claim.**
>
> We agree that evaluation on a larger model would further strengthen our claim. We therefore added results on Qwen2.5-32B: our method still provides consistent gains of 20.8\% over the base model while remaining 9.4\% better than MCMC. Our method also remains faster, requiring 0.36 minutes per prompt, compared with 4.1 minutes for MCMC.
>
> | Method | MATH500 | HumanEval | GPQA |
> |---|---:|---:|---:|
> | **Qwen2.5-32B** |  |  |  |
> | Base | 0.577 | 0.392 | 0.376 |
> | Low-temperature | 0.682 | 0.507 | 0.414 |
> | MCMC Power Sampling | 0.738 | 0.567 | 0.383 |
> | **Power Sampling (ours)** | **0.748** | **0.793** | **0.429** |
>
> **Table 1:** Evaluation on Qwen2.5-32B.
>
>
> > **Extending proof to the batch tokens extension.**
>
> We have now derived a bias bound for the batch tokens extension, which shows how the approximation error accumulates across block updates and how it depends on Top-$K_t$ and truncated lookahead $H_t$.
>
> In particular, for Top-$K_t$, we obtain
> $$
> \hat{p}\_{\alpha, \kappa}^{(\mathrm{pow})}(x\_t | \mathbf{q}, \mathbf{x}\_{0:t-1}) = \hat{p}\_{\alpha}^{(\mathrm{pow})}(x\_t | \mathbf{q}, \mathbf{x}\_{0:t-1}) + \mathcal{O}(\kappa),
> $$
>
> where $\kappa$ is the cumulative probability mass excluded by the Top-$K_t$ truncation. This error decreases as $K_t$ increases and vanishes when $K_t=\mathcal V$.
>
> For truncated lookahead, we obtain
>
> $$
> \hat{p}\_{\alpha, H\_t}^{(\mathrm{pow})}(x\_t | \mathbf{q}, \mathbf{x}\_{0:t-1}) = \hat{p}\_{\alpha}^{(\mathrm{pow})}(x\_t | \mathbf{q}, \mathbf{x}\_{0:t-1}) + \mathcal{O}\left(\sum\_{x\_t' \in \mathcal{V}} \eta\_{t, H\_t}\right),
> $$
>
> where $\eta_{t,H_t}$ measures the dependence of the sequence probability on context beyond horizon $H_t$, and decreases as $H_t$ increases. In the batch setting, the total bias across a block is bounded by the accumulation of these per-step approximation terms. We will include the full theorem and proof in the revised appendix.

---

> > ### Author Rebuttal · Reviewer_2SZZ · 2026-04-02
> >
> > I think that the current work does not yet demonstrate a significant advantage over reinforcement learning, and it has not been validated on mainstream large-scale models (e.g., 72B parameters). Nevertheless, this is a very interesting piece of work that proposes a valuable research direction. Therefore, I am inclined to increase my score and confidence.

---

> > > ### Author Response · Authors · 2026-04-04
> > >
> > > Thank you for reading our rebuttal and for increasing your score. We appreciate your thoughtful and positive feedback. Although we could not evaluate a 72B model during the rebuttal due to compute constraints, we will include the 32B results and further clarify our positioning with respect to reinforcement learning in the revised paper.

---

### Official Review · Reviewer_9XjT · 2026-03-12

**Soundness:** 3
**Presentation:** 3
**Significance:** 3
**Originality:** 3
**Overall Recommendation:** 4
**Confidence:** 3

**Summary:**

This paper proposes a training- and verifier-free approximation to power-distribution sampling for LLM reasoning that avoids costly MCMC. The core idea is an explicit token-level decomposition of the power distribution into a low-temperature factor multiplied by a future-quality scaling term that can be estimated with autoregressive Monte Carlo rollouts and debiased via a jackknife-baesd correction. Experiments across four 7B-scale models and three benchmarks (MATH500, HumanEval, GPQA) show accuracy competitive with, and sometimes better than, MCMC-based power sampling and one-shot GRPO, while achieving over 10× lower inference latency than MCMC.

**Compliance With Llm Reviewing Policy:**

Affirmed.

**Final Justification:**

My concerns were all fully addressed; therefore, I maintain my positive score.

**Key Questions For Authors:**

1. How does your method’s compute overhead compare to standard decoding in tokens/FLOPs, not just wall-clock? Please report tokens generated per prompt and total forward passes for a representative setting to complement Figure 2.

**Limitations:**

Yes

**Strengths And Weaknesses:**

## Strength
###  Novelty
- The paper formalizes a simple but useful decomposition of the global power distribution into a token-local term times a future-aware scalar, which turns a sequence-level objective into autoregressive sampling with per-token corrections.
- The use of autoregressive Monte Carlo to estimate the future-scaling factors and a jackknife correction to reduce ratio-estimator bias provides a clean, training-free, verifier-free alternative to MCMC power sampling.
- The connection links distribution sharpening to standard decoding primitives, potentially enabling broader adoption and systems integration.
### SOundness
- Evaluations span multiple tasks (math, code, science QA) and model families, including an RL-tuned model, which helps assess robustness and transfer.
- Comparisons include base decoding, low-temperature, Best-of-N, MCMC power sampling, and GRPO (Math), with both accuracy and latency reported.
 - Ablations are thorough -- performed on α, rollout budget M_t, and candidate set size K_t, plus Pass@K analysis.
### Presentation
- The problem setup and notation are largely clear; the decomposition (Theorem 3.1) and estimators are stated explicitly.
## Weaknesses
### Originality
- The central decomposition (Theorem 3.1) is an algebraic restatement of the definition of the power distribution via the chain rule; the conceptual novelty is modest, and the paper may overstate its theoretical depth.
- The practical algorithm relies on additional approximations (Top-K truncation of the normalization set, truncated lookahead H_t) whose induced bias/error are not analyzed; the theory does not bound these sources of error.
- The jackknife-corrected mixture of ratio estimators may yield negative probabilities before renormalization; the paper does not address numerical stability and projection back onto the simplex.
### Experimental methodologies' issues
- Compute accounting is incomplete: while speedups over MCMC are reported, overhead versus standard decoding (and absolute token-level compute) is not systematically quantified, making it hard to position the method on the broader inference-compute frontier.
- Variability is not reported (no confidence intervals or multiple runs), and it is unclear how $\alpha$ and other hyperparameters were tuned.
### Presentation clarity
- Key practical knobs (e.g., truncated horizon H_t, batch B) are deferred to the appendix; the main text lacks a principled discussion of how these affect compute and accuracy.
- The cost model is stated informally (O(sum_t K_t M_t) “generated sequences”) but does not translate this to token-level FLOPs or wall-clock under different sequence lengths.
### Missing related work or comparisons
- The paper briefly mentions inference-time planning/sampling methods but does not empirically compare to recent, competitive training-free approaches (ID-sampling; self-consistency) or discuss how its accuracy-latency tradeoffs relate to reported inference-scaling laws.

---

> ### Author Rebuttal · Authors · 2026-03-31
>
> We thank the reviewer for the positive feedback and thoughtful questions.
>
> > **Theorem 3.1 is an algebraic restatement**
>
> We agree that Theorem 3.1 is a careful derivation of power distributions, but we would like to clarify that this decomposition has not been made explicit in the LLM literature and it reveals power distributions of autoregressive LLMs as future-weighted low-temperature distributions, enabling our scalable approximation. To avoid overclaiming, we will soften the wording and add a proof sketch in the revised paper.
>
> > **Induced bias of Top-K and $H_t$.**
>
> We have derived new bounds quantifying the bias introduced by Top-$K_t$ and the truncation $H_t$, and will include them in the revised paper. Please refer to our rebuttal to Reviewer 2SZZ for more details.
>
> > **The jackknife estimators may yield negative probabilities**
>
> We have studied the conditions under which the jackknife term can become negative. This can occur only if, for a given token, exactly one sampled rollout has a probability that differs substantially from all the others, which is rare in LLMs. We examined this issue and did not observe it in our experiments. If such a case were to occur, a resampling step would address it. The full derivation will be presented in the updated paper due to space constraints.
>
> > **Hyperparameters tuning and how $H_t$ and $B$ affect compute and accuracy.**
>
> We followed the standard protocol for hyperparameter tuning: values were selected on a validation set and then fixed for the main experiments, while matching other setups in MCMC for fairness. We now provide an ablation over $H_t$ and $B$ on MATH in Qwen2.5-Math-7B (B=192 when sweeping $H$, and $H=192$ when sweeping $B$). We will add a more detailed discussion in the revision, and we note that since the average length is around 700, $B=768$ can generate full responses without lookahead in many questions, making its performance similar to low-temperature decoding.
>
> |  | Value | Accuracy (\%) | Time (min) |
> |---|---:|---:|---:|
> | **H_t** | 48  | 74.8 | 0.09 |
> |  | 96  | 75.6 | 0.16 |
> |  | 192 | 75.8 | 0.22 |
> |  | 384 | 76.2 | 0.29 |
> |  | 768 | 74.8 | 0.38 |
> | **B** | 48  | 76.4 | 0.43 |
> |  | 96  | 74.8 | 0.26 |
> |  | 192 | 75.8 | 0.22 |
> |  | 384 | 74.8 | 0.14 |
> |  | 768 | 70.4 | 0.11 |
>
> **Table 3**
>
> > **Variability is not reported**
>
> We will add a discussion of statistical reliability. We had time to make 5 runs on Qwen2.5-Math-7B for our method and low-temperature decoding, and the resulting variation is moderate: low-temperature decoding achieves $0.70 \pm 0.02$ on MATH, $0.30 \pm 0.02$ on GPQA, and $0.34 \pm 0.05$ on HumanEval; our method achieves $0.75 \pm 0.01$, $0.40 \pm 0.02$, and $0.59 \pm 0.02$, respectively.
>
> > **Empirically compare to recent training-free approaches and absolute token-level compute to position this method on the broader inference-compute frontier.**
>
> We added Self-Consistency as a baseline and will expand the related-work discussion to better position ID-sampling. Our method outperforms Self-Consistency@16 in 8/9 settings. Self-consistency can outperform MCMC on some tasks in Qwen2.5 models, and it even outperforms the GRPO result on MATH with Qwen2.5-Math-7B. However, it is less effective on the RL-tuned model, where top rollouts become similar and answer-level voting is less informative.
>
> For the compute question, we will report the average output lengths and total generated tokens in addition to wall-clock time in the revised paper. For a representative setting on MATH with Qwen2.5-Math-7B, the average output lengths are similar across methods: base decoding generates 640 tokens, compared with 667 for our method, 679 for MCMC, 682 for the GRPO model, and 690 for self-consistency. The total number of generated tokens during sampling for our method is 11k tokens and takes 0.22 min/prompt, compared with 12k tokens and 0.28 min/prompt for self-consistency. Our method achieves better accuracy with comparable sampling cost, while focusing on power distribution at the trajectory level rather than answer level. More generally, our method can also be used as the base strategy and then combined with self-consistency or BoN to improve diversity when answer-level aggregation is less effective.
>
> | Method | MATH500 | HumanEval | GPQA |
> |---|---:|---:|---:|
> | **Qwen2.5-7B** |  |  |  |
> | Base | 0.498 | 0.329 | 0.278 |
> | Self Consistency | 0.670 | 0.742 | 0.282 |
> | MCMC Power Sampling | 0.706 | 0.622 | 0.318 |
> | Power Sampling (ours) | **0.708** | **0.756** | **0.349** |
> | **Qwen2.5-Math-7B** |  |  |  |
> | Base | 0.496 | 0.329 | 0.278 |
> | Self Consistency | **0.789** | 0.591 | 0.384 |
> | MCMC Power Sampling | 0.748 | 0.573 | 0.389 |
> | Power Sampling (ours)| 0.758 | **0.604** | **0.409** |
> | **DeepSeek-Math-7B-RL** |  |  |  |
> | Base | 0.492 | 0.524 | 0.333 |
> | Self Consistency | 0.492 | 0.524 | 0.349 |
> | MCMC Power Sampling | 0.494 | 0.530 | 0.349 |
> | Power Sampling (ours) | **0.502** | **0.549** | **0.364** |
>
> **Table 4**

---

> > ### Author Rebuttal · Reviewer_9XjT · 2026-04-04
> >
> > Thank you for the detailed rebuttal. I will maintain my positive score.

---

> > > ### Author Response · Authors · 2026-04-04
> > >
> > > Thank you for reading our rebuttal and for your thoughtful feedback and questions. We are glad that our responses addressed your concerns, and we will incorporate these points into the revised paper.

---

### Official Review · Reviewer_Lr9B · 2026-03-13

**Soundness:** 3
**Presentation:** 3
**Significance:** 3
**Originality:** 3
**Overall Recommendation:** 5
**Confidence:** 4

**Summary:**

The authors develop an efficient approximation to MCMC algorithms that target sampling from the global power distribution of an autoregressive model. Instead of employing MCMC, the authors introduce a theoretically grounded approximator for the next-token marginals of the power distribution using a collection of rollouts. The resulting LLM-sampler matches, if not outperforms, the MCMC counterpart while maintaining significantly lower inference latency.

**Compliance With Llm Reviewing Policy:**

Affirmed.

**Final Justification:**

My concerns were all adequately addressed, and I maintain my original positive score.

**Key Questions For Authors:**

- In Figure 6, it appears as if occasionally, accuracy drops as the number of rollout samples increases. In general, is there a nice increase in performance with scaling the number of rollout samples?

- Does performance noticeably decrease by omitting the jackknife correction (i.e. simply using the naive lookahead rollout estimator)?

- Could inference latency be further decreased by having a small (distilled model) perform the lookahead rollouts?

**Limitations:**

yes

**Strengths And Weaknesses:**

Strengths:

- The method significantly improves MCMC latency for power sampling while also improving its performance on a variety of benchmarks and models.

- The paper effectively employs well-motivated classic techniques for statistical bias correction that are also well supported theoretically and provides convergence results to the ideal power distribution.

Weaknesses:

- Would be interesting to see demonstrations on larger models especially given the reduced inference cost, but understandable given the difficulty in scaling the given baselines to these larger models.

---

> ### Author Rebuttal · Authors · 2026-03-31
>
> We thank the reviewer for the positive assessment and for the helpful suggestions for improving this work.
>
> > **Would be interesting to see demonstrations on larger models**
>
> We agree that evaluation on a larger model would further strengthen our claim. We therefore added results on Qwen2.5-32B: our method still provides consistent gains of 20.8\% over the base model while remaining 9.4\% better than MCMC. Our method also remains faster, requiring 0.36 minutes per prompt, compared with 4.1 minutes for MCMC.
>
> | Method | MATH500 | HumanEval | GPQA |
> |---|---:|---:|---:|
> | **Qwen2.5-32B** |  |  |  |
> | Base | 0.577 | 0.392 | 0.376 |
> | Low-temperature | 0.682 | 0.507 | 0.414 |
> | MCMC Power Sampling | 0.738 | 0.567 | 0.383 |
> | **Power Sampling (ours)** | **0.748** | **0.793** | **0.429** |
>
> **Table 1:**  Evaluation on Qwen2.5-32B.
>
> > **In general, is there a nice increase in performance with scaling the number of rollout samples?**
>
> Increasing the number of rollout samples reduces the variance of the estimator, but task accuracy is not guaranteed to improve monotonically. As the number of rollouts increases, the estimator becomes less noisy and better approximates the target power distribution. Empirically, as shown in Figures 4 and 6, we observe clear improvement from very small rollout budgets, followed by saturation with occasional non-monotonic fluctuations.
>
> > **Does performance noticeably decrease by omitting the jackknife correction?**
>
> Yes, the jackknife estimator is especially helpful when the rollout budget $M_t$ is limited. To verify this empirically, we added an ablation comparing power sampling with the naive rollout estimator and the jackknife estimator in Table 2. Across the 9 model-task settings, while the naive estimator remains competitive in some settings, omitting the jackknife correction leads to an average drop of 4.29\%.
>
> | Method | MATH500 | HumanEval | GPQA |
> |---|---:|---:|---:|
> | **Qwen2.5-7B** |  |  |  |
> | Base | 0.498 | 0.329 | 0.278 |
> | Low-temperature | 0.628 | 0.524 | 0.303 |
> | MCMC Power Sampling | 0.706 | 0.622 | 0.318 |
> | **Power Sampling (without jackknife estimator)** | 0.650 | 0.677 | 0.318 |
> | **Power Sampling (ours)** | 0.708 | 0.756 | 0.349 |
> | **Qwen2.5-Math-7B** |  |  |  |
> | Base | 0.496 | 0.329 | 0.278 |
> | Low-temperature | 0.690 | 0.512 | 0.353 |
> | MCMC Power Sampling | 0.748 | 0.573 | 0.389 |
> | **Power Sampling (without jackknife estimator)** | 0.744 | 0.530 | 0.353 |
> | **Power Sampling (ours)** | 0.758 | 0.604 | 0.409 |
> | **DeepSeek-Math-7B** |  |  |  |
> | Base | 0.362 | 0.415 | 0.333 |
> | Low-temperature | 0.366 | 0.427 | 0.430 |
> | MCMC Power Sampling | 0.424 | 0.470 | 0.345 |
> | **Power Sampling (without jackknife estimator)** | 0.416 | 0.487 | 0.338 |
> | **Power Sampling (ours)** | 0.464 | 0.487 | 0.364 |
>
> **Table 2:** Performance comparison of Power Sampling with and without the jackknife estimator (i.e., with the naive lookahead).
>
> > **Could inference latency be further decreased by having a small (distilled model) perform the lookahead rollouts?**
>
> Thank you for your suggestion, and it is an exciting idea to use lookahead rollouts generated by a small distilled model to further improve the inference latency, which is also nicely connected to speculative decoding with verification. The quality-speedup tradeoff would depend on how close the small model distribution is to the base model. It is an interesting avenue for future research.

---

> > ### Author Rebuttal · Reviewer_Lr9B · 2026-04-02
> >
> > I thank the authors for their rebuttal. My questions have been addressed, and I maintain my original positive score.

---

> > > ### Author Response · Authors · 2026-04-04
> > >
> > > Thank you for reading our rebuttal and for your positive feedback and helpful suggestions for improving this work. We will be sure to include these points in our revised paper.

---

### Decision · Program_Chairs · 2026-04-30

**Decision:**

Accept (regular)

**Comment:**

This paper proposes an efficient, training-free and verifier-free method for approximating power-distribution sampling in large language models, which is commonly used to improve reasoning quality but typically relies on expensive MCMC procedures. Both theoretical analysis and empirical results support the effectiveness of the proposed method.

The reviewers generally agree that the paper presents a well-motivated and practically valuable approach to approximating power-distribution sampling without MCMC, achieving substantial reductions in inference latency while maintaining competitive or superior performance across multiple benchmarks. The proposed decomposition and use of autoregressive Monte Carlo with bias correction are viewed as technically sound and empirically effective, and the rebuttal successfully addressed major concerns, reinforcing reviewers’ positive assessments.

Overall, the paper is considered a solid and timely contribution for LLMs. Based on the reviewers’ consensus (3 accept and 1 weak accept) and the satisfactory rebuttal, the AC’s final decision is to accept the paper, while encouraging the authors to further strengthen it by incorporating additional results and discussions from the rebuttal into the final revision.